# Mitigating Coordinate Prediction Bias from Positional Encoding Failures

## Abstract

Multimodal large language models (MLLMs) excel at vision–language tasks such as VQA and document understanding, yet precise coordinate prediction remains challenging. High-resolution inputs exacerbate this difficulty by producing long token sequences that weaken positional encodings and introduce directional biases in coordinate outputs. We investigate this phenomenon by analyzing how MLLMs behave when visual positional encodings (VPEs) are deliberately perturbed through shuffling. Our analysis reveals that such perturbations induce predictable, non-random coordinate biases rather than random errors, suggesting that models rely on internal positional priors when spatial grounding signals are degraded. Crucially, we observe similar directional error patterns in natural high-resolution datasets, indicating that positional encoding failures are a key bottleneck for accurate coordinate prediction at scale. To address this issue, we propose Vision-PE Shuffle Guidance (VPSG), a training-free test-time method that leverages the directional nature of these biases for correction. VPSG runs auxiliary decoding with shuffled VPEs to isolate position-unconditioned tendencies, then uses this as negative evidence to guide digit prediction while preserving coordinate format through a lightweight finite-state machine. Experiments on ScreenSpot-Pro demonstrate reliable improvements, highlighting positional encoding robustness as a critical factor for spatial reasoning in MLLMs.

## 1 Introduction

Recent advances in large language models (LLMs; Touvron et al. 2023a; Chiang et al. 2023; Almazrouei et al. 2023; MosaicML 2023; Touvron et al. 2023b; OpenAI 2022; Google 2023) have improved language understanding and generation, but their text-only I/O limits perceptual and interactive use. Multi-modal LLMs (MLLMs) combine vision and text—e.g., Flamingo (Alayrac et al., 2022), Gemini (Team et al., 2023), and Qwen-VL (Bai et al., 2023; Wang et al., 2024; Bai et al., 2025)—to enable tasks such as visual QA, captioning, and document understanding. Coordinate prediction supports applications like object manipulation and GUI automation, where an MLLM outputs a 2-D point or bounding box (e.g., "[1000,500]"). High-resolution inputs make this harder: token and compute costs rise, and larger spatial extents increase pixel–patch misalignment, leading to coordinate drift (Li et al., 2025; Gao et al., 2024; Hsieh et al., 2024; Yen et al., 2024).

This coordinate drift primarily stems from the degradation of positional encodings at high resolutions, where conventional spatial anchors fail to scale reliably (Zhang et al., 2024). Positional encodings anchor visual tokens to image geometry and are crucial for coordinate prediction. VLMs typically apply (i) 2-D encodings in the vision encoder (e.g., ViT (Dosovitskiy et al., 2020)) and (ii) sequence-level schemes in the LLM (e.g., RoPE (Su et al., 2024)). High-resolution inputs push models into a long-context regime where attention diffuses and fine-grained spatial cues weaken. Cropping-based solutions (Tao et al., 2025; Wu et al., 2025) require coarse pre-localization and risk losing global semantics. Methods that enhance positional encodings (Ge et al., 2024; Chen et al., 2025b; Heo et al., 2024) help coarse tasks (e.g., VQA) but remain inadequate for precise coordinate prediction, where small biases cause numerical errors.

For both humans and machines, coordinate prediction requires precise awareness of the spatial arrangement of elements within an image. When the positional encodings of vision tokens are disrupted, for instance, by dividing the image into patches and shuffling their order, humans typically

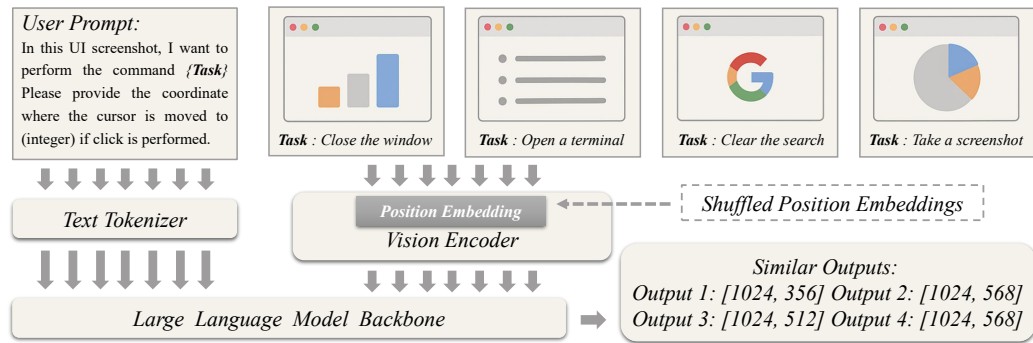

Figure 1: Effect of shuffling visual positional encodings: removing spatial conditioning causes the model to collapse to similar coordinate predictions across independent runs, indicating a position-unconditioned, directional bias rather than random variation. We also observe a similar clustered error pattern on high-resolution images (without shuffling), consistent with position-encoding failures.

fail to recover the correct coordinates and resort to random guesses. In contrast, MLLMs exhibit different behavior.

Our experiments show that under such perturbations, the deviations in their outputs are not random but display *systematic directional biases*. Furthermore, we observe that a substantial fraction of errors in high-resolution image datasets follows a distribution similar to that seen in these perturbation experiments. This suggests that, in long-context scenarios induced by high-resolution image inputs, the weakening of positional information amplifies the model's inherent directional biases, which in turn degrade coordinate prediction accuracy as shown in Figure 1. In this work, we seek to mitigate such biases and reinforce the contribution of positional encodings, thereby improving the robustness of MLLMs on position-sensitive tasks.

We propose Vision-PE Shuffle Guidance (VPSG), a training-free test-time guidance scheme that probes failure modes online and suppresses them during generation using only the base model. We view decoding through conditional probabilities where the main route estimates a position-conditioned token distribution (given image, prompt, and valid positional encodings), while the auxiliary routes approximate a position-unconditioned reference by shuffling the visual positional encodings. VPSG fuses these two signals at test time: the conditional–unconditional contrast amplifies information that is consistent with correct positions and suppresses content that persists when positional cues are removed. Practically, at each step we adjust only digit tokens by boosting the position-conditioned evidence and down-weighting the position-free tendency, and a lightweight FSM leaves commas, spaces, and brackets untouched. Under greedy decoding this behaves like subtracting a scaled "negative score" on digits, thereby strengthening the influence of positional information and stabilizing $[x, y]$ outputs without any training or architectural changes.

Two key design choices make VPSG precise and stable. Rather than relying on a single PE-shuffled auxiliary route, VPSG aggregates multiple shuffled routes in log space (geometric mean), yielding a robust estimate of the position-unconditioned bias and stabilizing the negative-evidence signal across inputs. In addition, VPSG applies a position-aware coefficient schedule: the guidance weight starts high for the first digit of $x$, decays geometrically for subsequent digits, resets at the first digit of $y$, and then decays again. This concentrates correction on the most influential digits while avoiding over-regularization of later positions and preserving natural numeric formatting. Our contributions can be summarized as follows:

- **Positional fragility analysis.** We show that perturbing visual positional encodings (VPEs) induces directional, repeatable biases in MLLM coordinate prediction, with similar effects at high resolution—revealing a resolution-dependent failure mode.

- **Training-free guidance.** We propose Vision-PE Shuffle Guidance (VPSG), a model-agnostic test-time method that shuffles VPEs to form counterfactual routes and guides final-layer digit logits via a lightweight finite-state machine.

## 2 RELATED WORK

**Coordinate prediction with MLLMs** With the rise of multi-modal LLMs (MLLMs) (Bai et al., 2023; Wang et al., 2024; Bai et al., 2025; Alayrac et al., 2022; Team et al., 2023; Ma et al., 2023; Yang et al., 2023; Liu et al., 2023a; Li et al., 2023; Liu et al., 2023b; Xu et al., 2025), coordinate prediction has been reformulated as a language-driven grounding problem, where models output coordinates as discrete token sequences rather than continuous regression targets. A particularly relevant domain is graphical user interface (GUI) interaction. There are many works that propose datasets to evaluate the performance of large models on GUI-related tasks (Cheng et al., 2024; Wu et al., 2024; Li et al., 2025). Recent works explore grounding instructions in screenshots for automated operation, mobile app understanding, or agent-based UI navigation. Data-centric approaches such as ShowUI (Lin et al., 2024) and UGround (Gou et al., 2025) synthesize large-scale training datasets to support the learning of efficient GUI agent models. OmniParser (Lu et al., 2024) leverages auxiliary visual models to annotate the positions of interface elements, thereby improving the performance of GPT-4V in GUI agent benchmarks. WebGUM(Furuta et al., 2023) introduces a hierarchical planning framework that integrates LLMs with execution modules and perceptual grounding, enabling structured decision-making in web-based tasks.

**Visual position encoding** Recent works emphasize that visual position encoding is crucial for scaling vision and multi-modal transformers (Wang et al., 2025). Beyond absolute or relative embeddings, variable schemes such as V2PE (Ge et al., 2024) and PyPE (Chen et al., 2025b) improve robustness in long-context and hierarchical perception. In structured document tasks, DocLayLLM (Liao et al., 2025) shows that lightweight 2D markers enhance layout understanding. Analyses of RoPE (Heo et al., 2024) highlight its extrapolation benefits, while semantic-aware encodings (Chen et al., 2025a) adapt to perceptual similarity. More recent efforts generalize or reinterpret RoPE via Fourier analysis (FoPE, Hua et al. (2024)) or trainable commuting matrices (ComRoPE, Yu et al. (2025)), and study its interaction with pooling mechanisms (Lee et al., 2025). Qi et al. analyzes how disproportionately large vision embedding norms suppress positional encodings in Vision-Language Models, leading to spatial reasoning failures. These results collectively indicate that designing flexible and resolution-robust encodings is central to advancing vision–language models. There is currently a lack of discussion on tasks that require precise location information, such as coordinate prediction.

## 3 METHODOLOGY

In this section, we introduce Vision-PE Shuffle Guidance (VPSG), a training-free, test-time guidance that stabilizes coordinate outputs in multimodal LLMs. VPSG runs the base model once on the normal input and in parallel creates shuffle-guided auxiliary views by perturbing only the visual positional encodings; discrepancies across these views reveal spurious numeric tendencies. Using this as negative evidence, VPSG gently steers the main decoding on digit tokens (leaving non-digits untouched) and improves $[x, y]$ reliability without fine-tuning or architectural changes.

### 3.1 CAUSAL VIEW OF COORDINATE PREDICTION

At the core of our study lies the observation that coordinate prediction in multimodal LLMs is governed by a causal mechanism: the output depends simultaneously on position-conditioned signals (e.g., visual positional encodings) and position-unconditioned signals (e.g., default digit tendencies). Prior analyses largely overlook the influence of these position-independent inputs, treating prediction errors as random noise. In contrast, we hypothesize that the non-positional pathway can

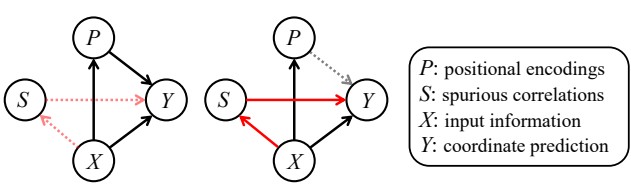

Figure 2: Causal view of coordinate prediction. Image content and prompt provide the intended causal effect on output digits (left). When VPEs are missing (e.g., at high resolution), the model relies on spurious correlations, leading to directional digit biases (right).

introduce directional bias when positional information is weak or missing. Motivated by this insight, we adopt a causal graph (Pearl et al., 2016; Pearl, 2018; Tang et al., 2020; Wang et al., 2022) as the analytical framework to explicitly model how position-related and position-free factors jointly shape the output and to identify spurious routes that degrade coordinate accuracy. Ideally, the output digits $[x, y]$ are determined jointly by the image content and the textual prompt, with visual positional encodings (VPEs) supplying spatial grounding. However, when VPEs are missing or unreliable—such as in high-resolution inputs—the causal pathway is disrupted. Lacking accurate positional cues, the model tends to rely on spurious correlations, for example overpredicting certain digits or repeating biased numeric patterns that are not grounded in the image.

As shown in Figure 2, we model coordinate prediction with a simple structural view over four nodes: input information $X$ (image content and prompt), positional encodings $P$ (visual spatial cues), spurious correlations $S$ (default numeric tendencies that emerge independently of the input, such as frequently repeated digits or preferred coordinate patterns), and the predicted coordinates $Y$. Here, $S$ captures position-unconditioned regularities in the model's training data or internal priors that can influence outputs even when visual evidence is weak or missing. A minimal Structural Causal Model is $Y = g(X, P, S)$, where $P$ is provided by the vision encoder (and varies with resolution), and $S$ summarizes non-causal numeric regularities the model can fall back on.

When $P$ is available and reliable, it supplies spatial grounding. The dominant pathways are $X \to Y$ and $P \to Y$. The influence of $S$ is negligible (dotted edges): although spurious patterns exist in the model's priors, they are largely blocked in practice because the model can rely on informative $X$ and $P$. When $P$ is absent or unreliable, the informative pathway weakens ($P \dashrightarrow Y$), and the spurious path without positional condition $S \to Y$ becomes comparatively strong. The model then defaults to biased numeric templates (e.g., over-predicting certain digits or repeating patterns) that are not supported by the input. In potential-outcome terms, the discrepancy $S(x) = Y(x, p_{\text{bad}}) - Y(x, p_{\text{good}})$ captures the shift in predictions attributable to the loss of positional grounding, with $p_{\text{good}}$ denoting a reliable PE setting and $p_{\text{bad}}$ an out-of-range or missing one.

This causal view clarifies the failure mode: positional degradation amplifies the non-causal route from $S$ to $Y$, yielding directional digit errors even when $X$ is unchanged. It also motivates our methodology: design a test-time procedure that (i) exposes the spurious route when $P$ is weak and (ii) suppresses its influence on the numeric tokens while preserving the informative flow from $X$ (and any usable $P$).

### 3.2 Bias Analysis

Given the causal graph in Figure 2, weakening the positional encodings $P$ increases the relative influence of spurious correlations $S$ on the output $Y$, distorting the digit distribution and pulling predictions away from the evidence in $X$. We therefore propose a two-part intervention: (i) expose the spurious route when $P$ is weak by constructing counterfactual views that differ only in positional cues, and (ii) suppress its impact on numeric tokens while preserving the informative flow from $X$ (and any usable $P$). Before presenting the intervention, we first establish empirically that the resulting errors are directional rather than random, confirming that the $S \to Y$ pathway is measurable.

Let $x$ be an input of size $(W_x, H_x)$ with diagonal $d_x = \sqrt{W_x^2 + H_x^2}$ to normalize scale across images. Under shuffled PEs, we run the model on $S$ cases and compute pairwise distances

$$d^{(i,j)}(x) = \left\| \hat{\mathbf{y}}^{(i)}(x) - \hat{\mathbf{y}}^{(j)}(x) \right\|_2, \qquad 1 \le i < j \le S, \tag{1}$$

then normalize as $\tilde{d}^{(i,j)}(x) = \frac{d^{(i,j)}(x)}{d_x}$. Pooling $\tilde{d}^{(i,j)}(x)$ over all inputs yields the shuffled-PE distance distribution $\mathcal{P}_{\text{shuffle}}$, while normal PEs give $\mathcal{P}_{\text{normal}}$. We compare their empirical means, against the scale-aware baseline $\mu_0 \approx 0.5214$ (Appendix D) to assess whether shuffled predictions collapse toward a small coordinate subset. Evidence of systematic, non-random bias is a clear left shift of $\mathcal{P}_{\text{shuffle}}$ toward zero relative to both $\mu_0$ (i.e., $\mathbb{E}[\tilde{d}] \ll \mu_0$) and $\mathcal{P}_{\text{normal}}$, indicating that predictions under shuffled PEs collapse to a few favored coordinates rather than dispersing as random fluctuations would.

Figure 3 summarizes the distance statistics. Across both Qwen2.5-VL-3B and Qwen2.5-VL-7B, the diagonal-normalized average pairwise distance under shuffled positional encodings is consistently small ($\tilde{d} \approx 0.16$), whereas the normal-PE condition exhibits substantially larger dispersion ($\tilde{d} \approx$

0.40–0.44). This substantial gap, far exceeding the baseline dispersion of random uniform points, confirms that when positional encodings are disrupted the model outputs collapse to a small set of preferred coordinates rather than spreading randomly. The consistent pattern across model scales demonstrates that the observed systematic directional bias is an inherent property of the architecture rather than a size-related artifact.

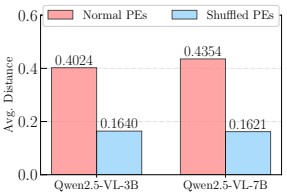 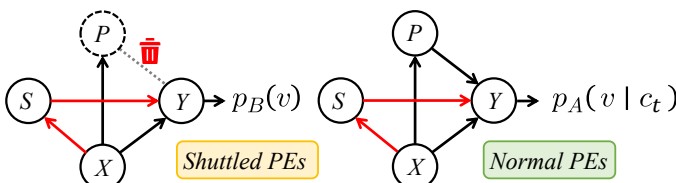

Figure 3: Diagonal-normalized average pairwise distance $\tilde{d}$ between coordinate predictions under *Normal PEs* and *Shuffled PEs*.

Figure 4: We obtain a position-*unconditioned* reference by shuffling visual positional encodings (left) and fuse it with the position-*conditioned* prediction (right). This conditional–unconditional contrast serves as negative evidence for digits, strengthening positional cues and suppressing spurious numeric patterns during decoding.

Based on the empirical and causal analysis of systematic bias caused by missing positional encodings presented above, we propose a bias-reduction strategy that remains unchanged during training but is inserted during inference: Vision-PE Shuffle Guidance (VPSG). Inspired by classifier-free guidance (CFG) (Ho & Salimans, 2022), we mitigate the impact of directional bias from the perspective of probability distribution.

**Overall algorithm.** VPSG runs one *main route* with normal visual positional encodings (PEs) and several *auxiliary routes* with randomly shuffled PEs. We let $c_t$ denote the position-conditioned context at step $t$, which primarily captures the positional encoding information from the visual encoder that provides spatial grounding for decoding. At each decoding step, we contrast the **position-conditioned** prediction $p_A(v \mid c_t)$ from the main route with an aggregated, **position-unconditioned** reference $p_B(v)$ formed by combining multiple shuffled routes. This contrast acts as *negative evidence* on digit tokens, while non-digit tokens (commas, spaces, brackets) remain untouched as shown in Figure 4. A finite-state machine (FSM) tracks whether the model is decoding the $x$ or $y$ coordinate and indicates when digit-specific guidance should be applied. This mechanism preserves the required $[x, y]$ format throughout decoding and prevents structural errors that could arise from spurious tokens or misaligned guidance.

**Proposition 1** (VPSG token guidance). *Let $\mathcal{D} \subset \mathcal{V}$ denote the digit subset of the vocabulary and $\alpha_t$ the step-wise guidance coefficient determined by the FSM. The VPSG-adjusted distribution satisfies*

$$p_{\text{VPSG}}(v \mid c_t) \propto \begin{cases} \exp\big(\log p_A(v \mid c_t) - \alpha_t \tilde{\ell}_B(v)\big), & v \in \mathcal{D}, \\ p_A(v \mid c_t), & v \notin \mathcal{D}, \end{cases} \tag{2}$$

*where $\tilde{\ell}_B(v) = \log p_B(v)$ is the log-probability of the position-unconditioned reference.*

This compact formula shows that VPSG subtracts a scaled log-probability from the digit logits of the main route while leaving non-digit tokens unchanged. The aggregation of $p_B(v)$ and the scheduling of $\alpha_t$ are described below; the proof that this form is equivalent to the classifier-free guidance view is provided in Appendix B.

**Seeds aggregation.** The auxiliary reference $p_B(v)$ is estimated by running the model on $S$ independent PE-shuffled seeds and aggregating their log-probabilities,

$$\tilde{\ell}_B(v) = \mathbb{E}_{\sigma \sim \nu}\Big[\log p_B^{(\sigma)}(v)\Big] = \int_{\sigma \in \mathcal{S}} \log p_B^{(\sigma)}(v)\, d\nu(\sigma) \approx \frac{1}{S} \sum_{s=1}^{S} \log p_B^{(s)}(v), \tag{3}$$

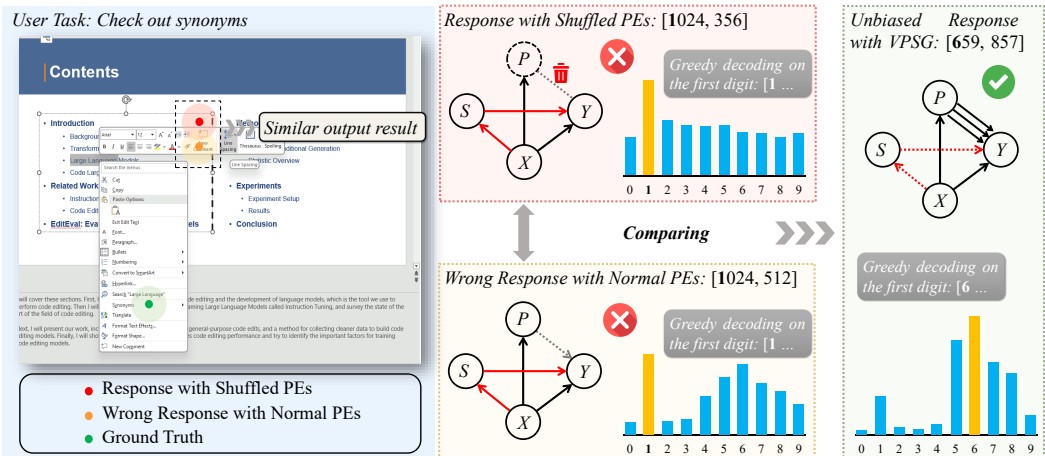

Figure 5: Qualitative example of VPSG on a Screenspot-Pro case. The base model with normal positional encodings produces a biased coordinate prediction ([1024, 512]), while the same model with shuffled positional encodings collapses to a similar but consistently biased point ([1024, 356]), revealing a directional position-unconditioned tendency. Applying VPSG corrects this bias and outputs the accurate ground truth ([659, 857]) by integrating negative evidence from multiple shuffled runs and reweighting digit logits, demonstrating how VPSG suppresses spurious patterns and restores faithful spatial grounding.

where $\sigma$ indexes shuffle transformations and $\nu$ is the (uniform) seed measure. This log-space (geometric mean) aggregation provides a robust Monte-Carlo estimate of the position-unconditioned bias prior.

**Coefficient decay.** To focus correction on the most influential digits, VPSG uses a geometric decay on $\alpha_t$ along the decoding sequence. Let $k_x$ (resp. $k_y$) be the index of the current digit within $x$ (resp. $y$). The guidance coefficient is scheduled as

$$\alpha_t = \begin{cases} \alpha \, \mathrm{decay}^{k_x-1}, & \text{decoding the } k_x\text{-th digit of } x, \\ \alpha, & \text{first digit of } y, \\ \alpha \, \mathrm{decay}^{k_y-1}, & \text{decoding the } k_y\text{-th digit of } y, \end{cases} \tag{4}$$

where $0 < \mathrm{decay} < 1$. This schedule emphasizes the most significant digits, resets at the first $y$ digit, and then tapers off, preventing over-regularization on later positions.

**Summary.** Overall, VPSG addresses this by effectively "asking twice": once with the normal input and again with a version whose visual positions are shuffled. Disagreements from the shuffled view act as negative evidence, gently steering the model's digit choices back toward what the image supports, while leaving non-digit text untouched. The result is a simple, plug-in procedure at inference time that stabilizes $[x, y]$ predictions without changing training or model architecture. It is model-agnostic, requires no retraining, and exactly recovers the baseline behavior when $\alpha_t \to 0$. The complete VPSG algorithm is shown in Appendix C. As shown in Figure 5, we provide a qualitative example of VPSG on a Screenspot-Pro case.

## 4 EXPERIMENTS

In this section, we evaluate the performance of VPSG when applied to VLLMs. As a plug-and-play, training-free approach, VPSG enhances existing models on coordinate prediction tasks, providing consistent improvements without requiring additional fine-tuning or architectural modifications.

### 4.1 EXPERIMENTAL SETTINGS

**Datasets.** We adopt the widely used ScreenSpot-Pro dataset to evaluate the performance of our method. ScreenSpot-Pro is a recently released benchmark for GUI grounding, consisting of real high-resolution desktop screenshots spanning 23 applications (e.g., VSCode, Photoshop, AutoCAD), five industry categories, and three operating systems, with precise annotations provided by professional users as shown in Table 1. The dataset is particularly challenging because target UI elements are often extremely small, occupying on average only 0.07% of the screen area. We evaluate VPSG under this realistic, high-resolution, and difficult setting to validate its effectiveness in improving localization accuracy.

**Models.** We adopt Qwen2.5-VL (Bai et al., 2025) as our test model, including configurations with 3B and 7B parameters. Unlike previous multimodal models, Qwen2.5-VL can directly output absolute coordinates for grounding tasks without requiring additional post-hoc alignment. The Qwen2.5-VL series has demonstrated strong performance on standard coordinate prediction benchmarks, even surpassing some specialized GUI models. However, its performance still degrades considerably in high-resolution scenarios, highlighting the inherent difficulty of precise localization under long-context inputs.

Table 1: Category and UI-type counts in Screenspot-Pro.

| Group | Text | Icon | Total |
|---|---|---|---|
| CAD | 197 | 64 | 261 |
| Creative | 198 | 143 | 341 |
| Dev | 154 | 145 | 299 |
| OS | 107 | 89 | 196 |
| Office | 177 | 53 | 230 |
| Scientific | 144 | 110 | 254 |
| **All** | **977** | **604** | **1581** |

**Method configurations.** For all evaluated models, we adopt greedy decoding to eliminate randomness and ensure reproducibility. For our proposed method VPSG, we identify the optimal hyperparameter configuration through grid search, with $\alpha = 0.55$ and decay $= 0.4$.

**Compared models** We evaluate a broad spectrum of multimodal models with an emphasis on general-purpose and training-free baselines, which are particularly important for assessing the effectiveness of our method without additional fine-tuning. This group includes Qwen-VL-7B (Bai et al., 2023), GPT-4o (Achiam et al., 2023), Qwen2-VL-7B (Wang et al., 2024), and MiniCPM-V, representing strong generalist vision–language models that can directly perform coordinate prediction. We further include the recent Qwen2.5-VL family (3B and 7B), which serves as our primary base model for applying VPSG and can output absolute coordinates without post-hoc alignment. For completeness, we also report results of specialized GUI action models such as SeeClick (Cheng et al., 2024), OS-Atlas-4B/7B (Wu et al., 2024), ShowUI-2B (Lin et al., 2024), CogAgent (Hong et al., 2024), Aria-GUI (Yang et al., 2024), and UGround-7B (Gou et al., 2025), which are trained or instruction-tuned specifically for interface grounding tasks. This diverse set of baselines enables a comprehensive evaluation of VPSG across both generic and domain-specific settings. The experimental results of the above model are cited from Li et al. (2025).

### 4.2 OVERALL PERFORMANCE

As shown in Table 2, VPSG consistently improves both base models on the Screenspot-Pro benchmark when measured by percentage correct. On Qwen2.5-VL-3B, the overall percentage correct increases from 11.6 to 13.3, a gain of 1.7 percentage points. Clear improvements appear in multiple text-oriented categories, including Development (Text) from 18.8 to 24.7 (+5.9 points), Creative (Text) from 16.7 to 20.2 (+3.5 points), CAD (Text) from 8.1 to 10.2 (+2.1 points), Office (Text) from 24.3 to 26.6 (+2.3 points)), and Scientific (Text) from 20.8 to 21.5 (+0.7 points). Several icon-oriented settings also benefit; for example, Development (Icon) rises from 1.4 to 2.1 (+0.7 points), Office (Icon) rises from 1.9 to 5.7 (+3.8 points) indicating that mitigating position-induced bias can stabilize landmark selection even when the target is an icon rather than text.

For Qwen2.5-VL-7B, the overall percentage correct increases from 18.5 to 19.1 (+0.6 points). Notable gains include Development (Text) from 37.7 to 40.9 (+3.2 points) and Office (Text) from 41.8 to 43.5 (+1.7 points), along with improvements in icon-oriented cases such as Office (Icon) from 11.3 to 13.2 (+1.9 points). Taken together, these results show that test-time negative-evidence guidance yields reliable lifts across model scales and interaction modes, enhancing both text-oriented

Table 2: Screen-based grounding results on SCREENSPOT-PRO. Each column reports the evaluation score (higher is better) for a category (*Development*, *Creative*, *CAD*, *Scientific*, *Office*, *OS*) split by UI type (*Text*/*Icon*); *Avg* is the unweighted mean across all columns. Rows marked "+ VPSG" apply our test-time guidance to the same base model, isolating the effect of the method. The best result among generalist models or training-free methods are highlighted in **bold** font.

| Model | Development | | Creative | | CAD | | Scientific | | Office | | OS | | Avg |
|---|---|---|---|---|---|---|---|---|---|---|---|---|---|
| | Text | Icon | Text | Icon | Text | Icon | Text | Icon | Text | Icon | Text | Icon | |
| **Trained GUI Action Models** 🔥 | | | | | | | | | | | | | |
| SeeClick | 0.6 | 0.0 | 1.0 | 0.0 | 2.5 | 0.0 | 3.5 | 0.0 | 1.1 | 0.0 | 2.8 | 0.0 | 1.1 |
| OS-Atlas-4B | 7.1 | 0.0 | 3.0 | 1.4 | 2.0 | 0.0 | 9.0 | 5.5 | 5.1 | 3.8 | 5.6 | 0.0 | 3.7 |
| ShowUI-2B | 16.9 | 1.4 | 9.1 | 0.0 | 2.5 | 0.0 | 13.2 | 7.3 | 15.3 | 7.5 | 10.3 | 2.2 | 7.7 |
| CogAgent-18B | 14.9 | 0.7 | 9.6 | 0.0 | 7.1 | 3.1 | 22.2 | 1.8 | 13.0 | 0.0 | 5.6 | 0.0 | 7.7 |
| Aria-GUI | 16.2 | 0.0 | 23.7 | 2.1 | 7.6 | 1.6 | 27.1 | 6.4 | 20.3 | 1.9 | 4.7 | 0.0 | 11.3 |
| UGround-7B | 26.6 | 2.1 | 27.3 | 2.8 | 14.2 | 1.6 | 31.9 | 2.7 | 31.6 | 11.3 | 17.8 | 0.0 | 16.5 |
| OS-Atlas-7B | 33.1 | 1.4 | 28.8 | 2.8 | 12.2 | 4.7 | 37.5 | 7.3 | 33.9 | 5.7 | 27.1 | 4.5 | 18.9 |
| **Generalist Models or Training-free Methods** ❄️ | | | | | | | | | | | | | |
| Qwen-VL-7B | 0.0 | 0.0 | 0.0 | 0.0 | 0.0 | 0.0 | 0.7 | 0.0 | 0.0 | 0.0 | 0.0 | 0.0 | 0.1 |
| GPT-4o | 1.3 | 0.0 | 1.0 | 0.0 | 2.0 | 0.0 | 2.1 | 0.0 | 1.1 | 0.0 | 0.0 | 0.0 | 0.8 |
| Qwen2-VL-7B | 2.6 | 0.0 | 1.5 | 0.0 | 0.5 | 0.0 | 6.3 | 0.0 | 3.4 | 1.9 | 0.9 | 0.0 | 1.6 |
| MiniCPM-V | 7.1 | 0.0 | 2.0 | 0.0 | 4.1 | **1.6** | 8.3 | 0.0 | 2.8 | 3.8 | 3.7 | 1.1 | 3.0 |
| Qwen2.5-VL-3B | 18.8 | 1.4 | 16.7 | 1.4 | 8.1 | **1.6** | 20.8 | 5.5 | 24.3 | 1.9 | 16.8 | 3.4 | 11.6 |
| Qwen2.5-VL-3B + VPSG | 24.7 | 2.1 | **20.2** | 2.1 | **10.2** | **1.6** | 21.5 | 5.5 | 26.6 | 5.7 | 15.9 | 1.1 | 13.3 |
| Qwen2.5-VL-7B | 37.7 | **2.8** | 19.7 | 2.1 | 7.6 | **1.6** | **31.3** | 5.5 | 41.8 | 11.3 | 29.9 | **10.1** | 18.5 |
| Qwen2.5-VL-7B + VPSG | **40.9** | 2.1 | 19.8 | **2.8** | 8.1 | **1.6** | 30.6 | **5.6** | **43.5** | **13.2** | 29.9 | **10.1** | **19.1** |

Table 3: Ablation study results. The analyzed method components include: (i) *seeds aggregation*, the robust log-space aggregation of multiple PE-shuffled auxiliary routes (*w/o seeds aggregation*: use a single seed, no aggregation); and (ii) *coefficient decay*, the position-aware geometric decay of the digit-only guidance weight with a reset at the first $y$ digit (*w/o coefficient decay*: use a constant guidance weight across digits).

**Qwen2.5-VL-3B**

| Setting | Avg | $\Delta$ |
|---|---|---|
| VPSG | 13.3 | – |
| w/o Seeds aggregation | 13.0 | ↓ 0.3 |
| w/o Coefficient decay | 11.9 | ↓ 1.4 |

**Qwen2.5-VL-7B**

| Setting | Avg | $\Delta$ |
|---|---|---|
| VPSG | 19.1 | – |
| w/o Seeds aggregation | 18.6 | ↓ 0.5 |
| w/o Coefficient decay | 18.2 | ↓ 0.9 |

and icon-oriented behaviors by suppressing spurious effects that emerge when positional signals are unreliable.

Our results underscore that a causal analysis of error pathways is essential for effectively mitigating coordinate prediction bias. By explicitly contrasting a position-conditioned distribution—obtained from normal positional encodings—with a position-unconditioned reference—derived from shuffled positional encodings—VPSG highlights and strengthens the influence of positional information in the final token distribution, while suppressing the position-agnostic tendencies that drive systematic, directional errors.

This comparison clarifies how positional cues causally affect output coordinates and ensures that guidance is grounded in a measurable contrast rather than heuristic adjustment. Because the intervention operates solely on the final-layer logits at test time, it remains fully compatible with pretrained MLLMs and introduces no additional training cost, architectural changes, or data requirements. Consequently, VPSG serves as a model-agnostic, plug-in method applicable to a broad range of coordinate-prediction tasks, enabling consistent and reproducible improvements across datasets and resolutions without modifying the existing training pipeline.

### 4.3 ABLATION STUDY

We perform ablations on Screenspot-Pro (percentage correct) to quantify the contribution of each VPSG component while holding all other settings fixed (same base model and decoding strategy).

#### 4.3.1 SEEDS AGGREGATION

The first component is seeds aggregation: instead of relying on a single PE-shuffled auxiliary route, the full method aggregates multiple routes in log space (geometric mean). Removing this component (w/o seeds aggregation) leads to a drop in performance. The rationale is that, although the errors induced by missing positional encodings are directional, any single random shuffle yields only a noisy sample from the underlying bias distribution and may not be representative. Aggregating across multiple seeds provides a more faithful estimate of the expectation of this position-unconditioned bias prior. This multi-path aggregation better recovers the output distribution absent positional conditioning.

#### 4.3.2 COEFFICIENT DECAY

The second key component of VPSG is coefficient decay. Table 3 highlights the importance of this design: removing coefficient decay (w/o coefficient decay) reduces the average percentage correct from $13.3 \rightarrow 11.9$ ($\downarrow 1.4$) on Qwen2.5-VL-3B and from $19.1 \rightarrow 18.2$ ($\downarrow 0.9$) on Qwen2.5-VL-7B. These drops are substantially larger than those caused by removing seeds aggregation, underscoring that position-aware scheduling is a primary driver of VPSG's gains. Beyond its positional-error weighting, coefficient decay also compensates for confidence attenuation along the digit sequence. Empirically, we observe that when the model predicts a multi-digit coordinate such as $[1234, 567]$, the confidence (logit margin between the top token and the runner-up) for the first digit is typically higher than for later digits: This progressive narrowing of logit margins indicates that later tokens are intrinsically more ambiguous, making them more sensitive to over-regularization. Applying a constant guidance coefficient would over-penalize these low-confidence positions, potentially distorting fine-scale digits or even the $[x, y]$ template.

By geometrically decaying $\alpha_t$ and resetting at the start of the $y$ coordinate, VPSG aligns the guidance strength with both positional importance and intrinsic confidence: it strongly constrains the high-order digits that dominate absolute error, while reducing the weight where the model's own uncertainty is higher and logit gaps are small. This targeted scheduling suppresses position-unconditioned biases without compromising the natural fine-grained structure of the output.

Taken together, these analyses confirm that coefficient decay is essential for balancing guidance strength with positional and confidence-based considerations, enabling VPSG to suppress position-unconditioned biases effectively.

## 5 CONCLUSION

We presented Vision-PE Shuffle Guidance (VPSG), a training-free and model-agnostic test-time method to improve coordinate prediction in multimodal large language models. Through a causal analysis of positional encodings, we showed that high-resolution inputs or perturbed visual positional embeddings induce systematic directional, position-unconditioned biases that cannot be eliminated by standard decoding. VPSG addresses this issue by running auxiliary decoding routes with shuffled positional encodings, using their outputs as negative evidence to suppress spurious numeric patterns while leaving non-digit tokens untouched. Key design elements such as multi-seed aggregation and position-aware coefficient decay—validated by ablation studies—ensure stable guidance that adapts to the natural confidence hierarchy of digit sequences. Extensive experiments on the ScreenSpot-Pro benchmark demonstrate consistent gains across model scales, including the strong Qwen2.5-VL series, without any fine-tuning or architectural changes. Our findings highlight the critical role of robust positional encoding for fine-grained spatial reasoning, and suggest that VPSG can serve as a practical plug-in for a wide range of grounding and coordinate-sensitive tasks in future vision–language systems.

ETHICS STATEMENT

This work does not involve human subjects, sensitive data, animal experiments, or any other aspect that raises ethical concerns. No potential risks of misuse or negative societal impact have been identified.

REPRODUCIBILITY STATEMENT

We are committed to ensuring reproducibility of our results. All code, along with instructions for data preprocessing, model configuration, and evaluation, will be released upon publication to enable full replication of the experiments and results reported in this paper.

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

## A  USE OF LARGE LANGUAGE MODELS

Large language models (LLMs) were used solely for language polishing and minor editorial assistance (e.g., grammar, wording, and clarity). They were not involved in the conception of research ideas, design of experiments, data analysis, or interpretation of results. All scientific content, methods, and conclusions were developed independently by the authors.

## B  PROOF OF PROPOSITION (VPSG TOKEN GUIDANCE)

**Setup.**  Let $\mathcal{V}$ be the vocabulary and $\mathcal{D} \subset \mathcal{V}$ the digit subset. At decoding step $t$, denote by $p_A(v \mid c_t)$ the position-conditioned (main-route) distribution and by $p_B(v)$ the position-*un*conditioned reference obtained from PE-shuffled auxiliary routes. Write $\ell_A(v \mid c_t) = \log p_A(v \mid c_t)$ and $\tilde{\ell}_B(v) = \log p_B(v)$. We assume: (i) the same logits processors are applied to both routes before any guidance; (ii) guidance acts only at the final-layer logits; and (iii) decoding is greedy, i.e., depends on the $\arg\max$ over logits.

**CFG form.**  Consider the (digit-only) classifier-free guidance (CFG) mixing:

$$p_{\mathrm{CFG}}(v \mid c_t) \ \propto \ \begin{cases} \dfrac{p_A(v \mid c_t)^{1+\lambda_t}}{p_B(v)^{\lambda_t}}, & v \in \mathcal{D}, \\ p_A(v \mid c_t), & v \notin \mathcal{D}, \end{cases} \tag{5}$$

with normalization constant $Z_{\mathrm{CFG}}(c_t)$ implicit over $v \in \mathcal{V}$. Taking logs for $v \in \mathcal{D}$,

$$\log p_{\mathrm{CFG}}(v \mid c_t) = (1 + \lambda_t)\,\ell_A(v \mid c_t) - \lambda_t\,\tilde{\ell}_B(v) - \log Z_{\mathrm{CFG}}(c_t). \tag{6}$$

**Digit-only positive affine rescaling.**  Define, for $v \in \mathcal{D}$,

$$\psi_t(v) \ = \ \frac{1}{1 + \lambda_t}\Big( \log p_{\mathrm{CFG}}(v \mid c_t) + \log Z_{\mathrm{CFG}}(c_t) \Big) \ = \ \ell_A(v \mid c_t) \ - \ \underbrace{\frac{\lambda_t}{1 + \lambda_t}}_{\alpha_t}\,\tilde{\ell}_B(v). \tag{7}$$

For $v \notin \mathcal{D}$, keep $\psi_t(v) = \log p_A(v \mid c_t)$ (no change).

**Lemma 1** (Argmax invariance under positive affine transforms)**.**  *Let $S \subseteq \mathcal{V}$ and $a > 0$, $b \in \mathbb{R}$. For any scores $\{u(v)\}_{v \in S}$, $\arg\max_{v \in S} u(v) = \arg\max_{v \in S}\{a\,u(v) + b\}$.*

*Proof.*  For $a > 0$, $u(v_1) \geq u(v_2) \iff a\,u(v_1) + b \geq a\,u(v_2) + b$. $\qquad\square$

Applying the lemma to equation 6 with $a = \frac{1}{1+\lambda_t}$ and $b = \frac{\log Z_{\mathrm{CFG}}(c_t)}{1+\lambda_t}$ *restricted to* $v \in \mathcal{D}$ shows that

$$\arg\max_{v \in \mathcal{D}} \log p_{\mathrm{CFG}}(v \mid c_t) \ = \ \arg\max_{v \in \mathcal{D}} \psi_t(v) \ = \ \arg\max_{v \in \mathcal{D}} \big(\ell_A(v \mid c_t) - \alpha_t\,\tilde{\ell}_B(v)\big).$$

Since $p_{\mathrm{CFG}}(v \mid c_t) = p_A(v \mid c_t)$ for $v \notin \mathcal{D}$ by equation 5, the combined (digit/non-digit) $\arg\max$ under CFG equals that under the piecewise score

$$s(v) \ = \ \begin{cases} \ell_A(v \mid c_t) - \alpha_t\,\tilde{\ell}_B(v), & v \in \mathcal{D}, \\ \ell_A(v \mid c_t), & v \notin \mathcal{D}, \end{cases}$$

which is the VPSG "negative-evidence" scoring form. Moreover, equation 7 yields the exact parameter mapping

$$\boxed{\alpha_t = \frac{\lambda_t}{1 + \lambda_t}} \qquad \Longleftrightarrow \qquad \boxed{\lambda_t = \frac{\alpha_t}{1 - \alpha_t}}.$$

**Equivalence under greedy decoding.**  Greedy decoding selects $\hat{v}_t = \arg\max_{v \in \mathcal{V}} \log p_{\mathrm{CFG}}(v \mid c_t)$. By the lemma and the piecewise definition above, the same $\hat{v}_t$ is obtained by maximizing $s(v)$, because: (i) on digits we used a positive affine transform of $\log p_{\mathrm{CFG}}$; (ii) on non-digits the two forms coincide; and (iii) both are compared in the same joint candidate set $\mathcal{V}$. Therefore CFG equation 5 and VPSG scoring $s(v)$ are *decision-equivalent* under greedy decoding.

**Normalization and distributional form.**   If a normalized distribution is desired, define

$$p_{\text{VPSG}}(v \mid c_t) \;=\; \frac{\exp(s(v))}{\sum_{u \in \mathcal{V}} \exp(s(u))}.$$

This coincides with equation 5 up to the digit-only affine rescaling leading to the same $\arg\max$, and yields the proposition's statement:

$$p_{\text{VPSG}}(v \mid c_t) \propto \begin{cases} \exp\big(\ell_A(v \mid c_t) - \alpha_t \, \tilde{\ell}_B(v)\big), & v \in \mathcal{D}, \\ p_A(v \mid c_t), & v \notin \mathcal{D}. \end{cases}$$

**Remarks on assumptions.**   (1) *Same logits processors:* ensures that when $\alpha_t \to 0$ (or $\lambda_t \to 0$) the VPSG rule recovers the baseline exactly. (2) *Final-layer intervention:* guarantees that the affine transformation does not change any upstream normalization. (3) *Greedy decoding:* makes decision-equivalence depend only on $\arg\max$; for sampling or beam search, the same mapping holds at the score level, but selection statistics may also depend on temperature/length penalties (which can still be shared across routes).

$\square$

## C   ALGORITHM OF VPSG

The complete algorithm is shown in Algorithm 1.

## D   EXPECTED DISTANCE IN THE UNIT SQUARE

Claim. If $X = (X_1, X_2)$ and $Y = (Y_1, Y_2)$ are independent and uniformly distributed on $[0,1]^2$, then

$$\mathbb{E}\big[\|X - Y\|_2\big] = \frac{2 + \sqrt{2} + 5\ln(1 + \sqrt{2})}{15} \;\approx\; 0.521405433.$$

Proof. Let

$$U = |X_1 - Y_1|, \qquad V = |X_2 - Y_2|.$$

For $u, v \in [0,1]$, the joint density of $(U, V)$ is

$$f_{U,V}(u, v) = 4(1 - u)(1 - v),$$

since each marginal $U$ (and $V$) is triangular with density $f_U(u) = 2(1-u)$ and $U, V$ are independent. The Euclidean distance is $R = \sqrt{U^2 + V^2}$. Hence

$$\mathbb{E}[R] = \int_0^1 \int_0^1 \sqrt{u^2 + v^2}\, 4(1 - u)(1 - v)\, du\, dv.$$

Switch to polar coordinates on the first quadrant: $u = r\cos\theta$, $v = r\sin\theta$ with $\theta \in [0, \pi/2]$ and Jacobian $r\, dr\, d\theta$. The square boundary imposes

$$0 \le r \le r_{\max}(\theta) = \min\{1/\cos\theta,\ 1/\sin\theta\}.$$

Noting $\sqrt{u^2 + v^2} = r$ and $(1 - u)(1 - v) = (1 - r\cos\theta)(1 - r\sin\theta)$, we obtain

$$\mathbb{E}[R] = 4 \int_0^{\pi/2} \int_0^{r_{\max}(\theta)} (1 - r\cos\theta)(1 - r\sin\theta)\, r^2\, dr\, d\theta.$$

---

**Algorithm 1** Vision-PE Shuffle Guidance (VPSG)

---

**Require:** Image $I$, prompt $q$, base model $\mathcal{M}$, seeds $\{s_1, \ldots, s_S\}$, base coefficient $\alpha$, decay factor $0 < \text{decay} < 1$

**Ensure:** Coordinate prediction $\hat{y} = [x, y]$

1: **Main route (position-conditioned):**
2: Run $\mathcal{M}$ on $(I, q)$ with normal positional encodings (PEs) to obtain token distribution $p_A(v \mid c_t)$.
3: **Auxiliary routes (position-unconditioned):**
4: **for** each seed $s$ **do**
5:     Shuffle PEs and run $\mathcal{M}$ to get $p_B^{(s)}(v)$.
6: **end for**
7: Aggregate in log-space:

$$\tilde{\ell}_B(v) \leftarrow \frac{1}{S} \sum_{s=1}^{S} \log p_B^{(s)}(v).$$

8: **FSM state tracking:**
9: Use a finite-state machine aligned to the $[x, y]$ template to determine whether the model is decoding a digit in $x$ or $y$.
10: **Coefficient scheduling:**
11: **if** decoding the $k_x$-th digit of $x$ **then**
12:     $\alpha_t \leftarrow \alpha \cdot \text{decay}^{k_x - 1}$
13: **else if** decoding the first digit of $y$ **then**
14:     $\alpha_t \leftarrow \alpha$
15: **else if** decoding the $k_y$-th digit of $y$ **then**
16:     $\alpha_t \leftarrow \alpha \cdot \text{decay}^{k_y - 1}$
17: **end if**
18: **Negative-evidence scoring:**
19: **for** each token $v$ **do**
20:

$$s(v) \leftarrow \begin{cases} \log p_A(v \mid c_t) - \alpha_t \tilde{\ell}_B(v), & v \in \mathcal{D}, \\ \log p_A(v \mid c_t), & v \notin \mathcal{D}. \end{cases}$$

21: **end for**
22: **Token selection:**
23: Choose $\hat{v}_t \leftarrow \arg\max_v s(v)$, append to output, and advance FSM.
24: **Termination:**
25: Repeat Steps 3–6 until EOS. Decode tokens into coordinates $\hat{y}$.

---

Split at $\theta = \pi/4$, where $r_{\max}$ changes:

$$\mathbb{E}[R] = 4\int_0^{\pi/4}\int_0^{\sec\theta}(1-r\cos\theta)(1-r\sin\theta)\,r^2\,dr\,d\theta + 4\int_{\pi/4}^{\pi/2}\int_0^{\csc\theta}(1-r\cos\theta)(1-r\sin\theta)\,r^2\,dr\,d\theta.$$

For fixed $\theta$, expand and integrate in $r$:

$$\int_0^a \left(r^2 - r^3(\cos\theta + \sin\theta) + r^4\sin\theta\cos\theta\right)dr = \frac{a^3}{3} - \frac{\cos\theta + \sin\theta}{4}a^4 + \frac{\sin\theta\cos\theta}{5}a^5.$$

With $a = \sec\theta$ on $[0, \pi/4]$ and $a = \csc\theta$ on $[\pi/4, \pi/2]$, we simplify:

$$J_1(\theta) = \frac{\sec^3\theta}{3} - \frac{\cos\theta + \sin\theta}{4}\sec^4\theta + \frac{\sin\theta\cos\theta}{5}\sec^5\theta = \sec^3\theta\left(\frac{1}{12} - \frac{1}{20}\tan\theta\right),$$

$$J_2(\theta) = \frac{\csc^3\theta}{3} - \frac{\cos\theta + \sin\theta}{4}\csc^4\theta + \frac{\sin\theta\cos\theta}{5}\csc^5\theta = \csc^3\theta\left(\frac{1}{12} - \frac{1}{20}\cot\theta\right).$$

Therefore

$$\mathbb{E}[R] = 4\int_0^{\pi/4}\left(\frac{1}{12}\sec^3\theta - \frac{1}{20}\sec^3\theta\tan\theta\right)d\theta + 4\int_{\pi/4}^{\pi/2}\left(\frac{1}{12}\csc^3\theta - \frac{1}{20}\csc^3\theta\cot\theta\right)d\theta.$$

Use the antiderivatives

$$\int\sec^3\theta\,d\theta = \tfrac{1}{2}\big(\sec\theta\tan\theta + \ln(\sec\theta + \tan\theta)\big), \quad \int\sec^3\theta\tan\theta\,d\theta = \tfrac{1}{3}\sec^3\theta,$$

$$\int\csc^3\theta\,d\theta = \tfrac{1}{2}\big(-\csc\theta\cot\theta + \ln(\csc\theta - \cot\theta)\big), \quad \int\csc^3\theta\cot\theta\,d\theta = -\tfrac{1}{3}\csc^3\theta,$$

evaluate at the limits $\theta \in \{0, \pi/4, \pi/2\}$, and use $\ln(\sqrt{2} - 1) = -\ln(\sqrt{2} + 1)$. The two $\theta$-ranges contribute symmetrically, giving

$$\mathbb{E}[R] = \frac{1}{3}\big(\sqrt{2} + \ln(1 + \sqrt{2})\big) - \frac{2}{15}\big(2\sqrt{2} - 1\big) = \frac{2 + \sqrt{2} + 5\ln(1 + \sqrt{2})}{15}.$$

This completes the proof. $\square$

Remark (application to images). The constant $\mu_\square$ is the dispersion benchmark for a unit square. For arbitrary image sizes $(W, H)$, either (i) anisotropically rescale coordinates to $[0, 1]^2$ before computing distances and compare to $\mu_\square$, or (ii) form a per-image Monte-Carlo null by sampling i.i.d. uniform points from $[0, W] \times [0, H]$ to estimate the appropriate baseline for that aspect ratio.

# E  MORE EXPERIMENT RESULT

## E.1  DISTANCE ANALYSIS

More details of distance analysis results are shown in Table 4

Table 4: Comparison of mean pairwise Euclidean distances (diagonal-normalized) across image resolutions on SCREENSPOT-PRO. Normal PEs exhibit significantly larger within-input distances, while shuffled PEs lead to much tighter clustering and systematic collapse of coordinate predictions, demonstrating the impact of positional information on spatial diversity.

| Image size | Qwen2.5-VL-3B | | Qwen2.5-VL-7B | |
|---|---|---|---|---|
| | Normal | Shuffled | Normal | Shuffled |
| 1920×1080 | 0.123 | 0.202 | 0.211 | 0.128 |
| 2160×1440 | 0.391 | 0.146 | 0.437 | 0.159 |
| 2560×1440 | 0.392 | 0.171 | 0.436 | 0.180 |
| 2560×1600 | 0.357 | 0.072 | 0.348 | 0.048 |
| 2560×1664 | 0.428 | 0.139 | 0.471 | 0.211 |
| 2880×1800 | 0.379 | 0.119 | 0.462 | 0.111 |
| 2992×1870 | 0.463 | 0.175 | 0.347 | 0.061 |
| 3456×2160 | 0.471 | 0.122 | 0.348 | 0.081 |
| 3456×2234 | 0.457 | 0.169 | 0.466 | 0.106 |
| 3840×1080 | 0.298 | 0.227 | 0.287 | 0.140 |
| 3840×2160 | 0.468 | 0.117 | 0.504 | 0.146 |
| 5120×1440 | 0.435 | 0.133 | 0.404 | 0.058 |
| 5120×2880 | 0.399 | 0.168 | 0.408 | 0.052 |
| 6016×3384 | 0.341 | 0.117 | 0.335 | 0.057 |
| **Overall mean** | **0.402** | **0.164** | **0.435** | **0.162** |

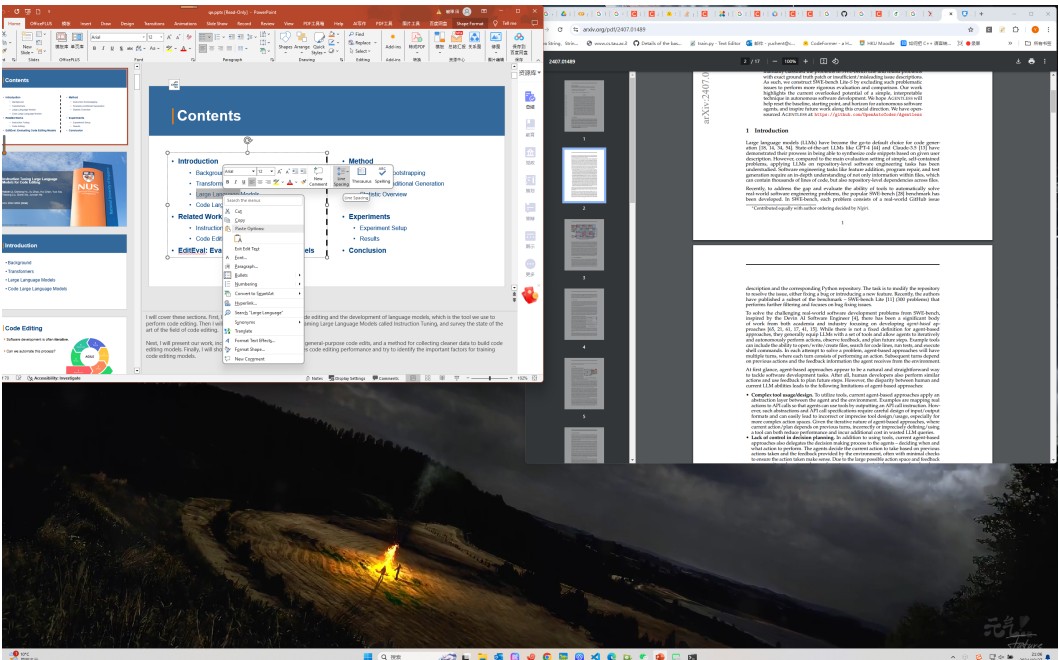

Figure 6: An image case from dataset ScreenSpot-Pro (ppt_windows/screenshot_2024-10-27_21-07-29.png)

### E.2 CASE STUDY WITH PER-STEP LOGITS.

To better illustrate how VPSG corrects coordinate prediction bias, we analyze a single case from SCREENSPOT-PRO (ppt_windows/screenshot_2024-10-27_21-07-29.png) Figure 6. The ground-truth bounding box center is $[659, 857]$. The base model without guidance predicts $[1024, 856]$, while VPSG successfully outputs the correct $[659, 857]$. Table 5 lists the top-10 logits probabilities at each decoding step. At the earliest $x$-digit steps, the uncorrected model shows a

Table 5: Per-step top-1 logits probabilities for each decoded token in a representative example. The base model without guidance drifts toward spurious large-$x$ digits (e.g., "1", "0" early), yielding an incorrect coordinate $[1024, 856]$. VPSG, by integrating negative evidence from multiple shuffled PE runs, suppresses these biased peaks and converges to the correct coordinate $[659, 857]$. This example highlights how VPSG stabilizes numeric decoding and restores faithful spatial grounding.

| Step | Token | VPSG Prob. | Base Prob. |
|------|-------|------------|------------|
| 1 | [ | 0.805 | 0.805 |
| 2 | 6 | 0.243 | 0.210 |
| 3 | 5 | 0.172 | 0.264 (0 highest) |
| 4 | 9 | 0.221 | 0.140 (2 highest) |
| 5 | , | 0.999 | 0.987 |
| 6 | space | 0.999 | 0.976 |
| 7 | 8 | 0.740 | 0.355 |
| 8 | 5 | 0.589 | 0.642 (but 6/3 confused) |
| 9 | 7 | 0.272 | 0.421 (6 highest) |
| 10 | ] | 0.999 | 0.999 |
| 11 | <eos> | 0.993 | 0.996 |

Table 6: Accuracy (%) of VPSG without seeds aggregation across categories and UI types. Both model sizes show clear drops compared with the full VPSG (see main text Table X): for instance, the 3B model drops from $13.3\%$ overall to $13.0\%$, and the 7B model from $19.1\%$ to $18.6\%$. Losses are consistent across text and icon settings, supporting the view that multi-seed aggregation provides a faithful estimate of the expected position-unconditioned bias and stabilizes the guidance effect.

| Category / UI type | 3B | | 7B | |
|--------------------|----------|----------|----------|----------|
| | Text (%) | Icon (%) | Text (%) | Icon (%) |
| CAD | 10.66 | 1.56 | 6.60 | 1.56 |
| Creative | 18.69 | 2.10 | 19.70 | 2.80 |
| Dev | 22.73 | 2.07 | 41.56 | 3.45 |
| OS | 15.89 | 2.25 | 28.97 | 12.36 |
| Office | 29.38 | 5.66 | 41.81 | 13.21 |
| Scientific | 20.14 | 2.73 | 27.78 | 4.55 |
| Overall | 19.55 | 2.48 | 26.71 | 5.46 |

strong bias toward larger numbers (e.g., tokens "1" and "0" dominate), reflecting spurious numeric priors induced by missing or unreliable positional encodings. VPSG integrates negative evidence from multiple shuffled PE runs and systematically downweights these spurious peaks, allowing the true digit sequence to emerge and stabilizing the final $[x, y]$ prediction.

### E.3 ABLATION: REMOVING SEEDS AGGREGATION.

To evaluate the contribution of seeds aggregation in VPSG, we remove the multi-seed log-space aggregation and instead rely on a single randomly shuffled positional encoding as the auxiliary route. Table 6 reports the detailed category- and type-level accuracies (percentage of correct predictions) for both Qwen2.5-VL-3B and Qwen2.5-VL-7B models. Without aggregation, accuracy drops across almost all groups and UI-types, confirming that a single random shuffle provides only a noisy sample of the underlying position-unconditioned bias distribution and cannot capture its full expectation. This validates the theoretical claim that multi-seed aggregation approximates the expected bias prior and yields more stable and accurate guidance.

### E.4 MORE DETAILS ABOUT BIAS ANALYSIS

Table 7 reports the ten most frequent individual numbers across all $[x, y]$ coordinate predictions. Normal PE = standard positional encodings; shuffled PE = visual positional encodings randomly

Table 7: Top-10 most frequent numbers appearing in prediction results for Qwen2.5-VL-3B and Qwen2.5-VL-7B under normal and shuffled positional encodings.

| Qwen2.5-VL-7B (normal PE) | | | Qwen2.5-VL-3B (normal PE) | | | Qwen2.5-VL-3B (shuffled PE) | | | Qwen2.5-VL-7B (shuffled PE) | | |
|---|---|---|---|---|---|---|---|---|---|---|---|
| Rank | Number | Freq | Rank | Number | Freq | Rank | Number | Freq | Rank | Number | Freq |
| 1 | 1024 | 296 | 1 | 1024 | 397 | 1 | 1024 | 591 | 1 | 1024 | 902 |
| 2 | 105 | 54 | 2 | 1056 | 82 | 2 | 1056 | 426 | 2 | 1234 | 582 |
| 3 | 10 | 35 | 3 | 356 | 73 | 3 | 568 | 184 | 3 | 567 | 580 |
| 4 | 2048 | 26 | 4 | 35 | 44 | 4 | 1000 | 182 | 4 | 672 | 270 |
| 5 | 2058 | 26 | 5 | 36 | 39 | 5 | 238 | 159 | 5 | 368 | 266 |
| 6 | 2016 | 25 | 6 | 10 | 36 | 6 | 512 | 141 | 6 | 384 | 162 |
| 7 | 1940 | 23 | 7 | 1234 | 28 | 7 | 560 | 141 | 7 | 200 | 41 |
| 8 | 100 | 23 | 8 | 105 | 27 | 8 | 1052 | 128 | 8 | 36 | 33 |
| 9 | 200 | 21 | 9 | 102 | 25 | 9 | 200 | 121 | 9 | 38 | 27 |
| 10 | 1056 | 20 | 10 | 1048 | 23 | 10 | 248 | 119 | 10 | 896 | 26 |

shuffled at inference time. All counts reflect total occurrences of a number as either the $x$ or $y$ component of predicted coordinates.

