# OpenReview forum: "Mitigating Coordinate Prediction Bias from Positional Encoding Failures"
_ICLR.cc/2026/Conference — ICLR 2026 Conference Withdrawn Submission_

### Official Review · Reviewer_WBAu · 2025-10-31

**Soundness:** 3
**Presentation:** 3
**Contribution:** 3
**Rating:** 4
**Confidence:** 3

**Summary:**

This paper improves the coordinate grounding capabilities of VLMs by reducing the impact of models' spurious visual position encoding (VPE) prior while reinforcing the influence of proper position encoding features. Through VPE shuffling experiments, the authors discovered that model exhibits a non random bias when the VPE is disturbed, suggesting that there is likely an underlying positional prior which causes a positional bias when model cannot rely on proper VPE. Therefore, the authors proposed VPSG, a training free decoding based method. VPSG requires models to perform inference multiple times, each with shuffled VPE at a different seed value. The aggregated shuffled VPE result is contrasted with proper VPE to obtain the final digit output during decoding process. Experimental results on Screenspot-Pro benchmark yields higher performance compared to the Qwen2.5VL base model.

**Strengths:**

Overall, the paper is fluent and the motivation for method design is well established. The authors reveal that randomly shuffling the VPE of a VLM leads to a non random bias, suggesting there is an inherent positional bias of models on top of the positional information obtained from VPE. This bias may lead to incorrect grounding of coordinates, especially when VPE becomes challenged in long context settings. The method design closely follows the discovery, which steers output generation towards VPE guided probability and away from inherent positional bias.

**Weaknesses:**

The authors disturb the position encoding of VLMs by shuffling operations and empirically prove that this shuffling led to preference of a few favoured coordinates. However, only shuffling operation is tested in this study. There might be other RoPE modification methods, such as NoPE (no position encoding), to find the built-in position bias.

The authors argue that position encoding failures would occur for long context high resolution grounding tasks. However, the base model of choice, Qwen2.5 VL, applies a 3D RoPE which already reduces the overall visual token distances and somewhat prevents long context visual position encodings. The validity of the method design can be further improved by showing evaluation results on VLMs that apply a 1D RoPE which is more prone to long context position encoding failures.

**Questions:**

1. Could you please provide more detail on how to shuffle PE? Do you shuffle the position index of vision tokens?
2. Could you provide more detail on how to derive the average pairwise distance as presented in figure 3?
3. Based on the experiment in figure 1, VPE shuffling should have little impact on the final coordinate grounding result. In that case, what is the accuracy of baseline mode evaluated on SCREENSPOT-PRO when PE is shuffled? Does it have values as compared to normal PE?
4. Is VPSG applicable to general grounding task? Currently it is only tested on SCREENSPOT-PRO. How does the method perform on other object grounding datasets such as RefCOCO?
5. Is VPSG applicable to other decoding methods? If so, could you provide some of the experimental results using different decoding strategies?
6. Since the model would run multiple inference passes, what is the inference speed in comparison to baseline models?

---

> ### Author Response · Authors · 2025-11-21
>
> We sincerely thank the reviewers for their constructive and insightful comments. We address each concern below. For clarity and readability, we restate each question in a concise and friendly form before responding.
>
> **Weakness 1**. The reviewer suggests that, beyond shuffling visual positional encodings, it may be beneficial to explore additional positional encoding modifications (such as removing positional encodings, i.e., NoPE) to further examine the presence of built-in positional biases in VLMs.
>
> R1.  We thank the reviewer for raising this thoughtful point and for recognizing the importance of analyzing built-in positional priors. The reviewer suggests exploring other positional encoding modifications (e.g., NoPE) to further reveal position-related biases. This is a valuable direction. However, we would like to clarify a potential misunderstanding regarding the scope and intention of our perturbation design.
>
> As mentioned in Sec. 3.2 (Bias Analysis) of our submission, our purpose is to introduce a causally controlled perturbation that (i) removes positional conditioning while (ii) keeping all visual token representations intact. Shuffling VPE indices satisfies these two requirements: it breaks geometric ordering without modifying token content, making the resulting counterfactual inference clean, disentangled, and easy to interpret.
>
> In contrast, altering the form of positional encoding, such as replacing RoPE with NoPE or zeroed embeddings, changes the model’s architecture-level positional encoding function. This requires full model re-training or re-alignment, which introduces substantial computational cost and departs from the 'training-free' nature of our method. Moreover, such modifications inject additional confounding factors: for instance, setting RoPE to zero removes representational geometry in a non-neutral way and may introduce new, unintended biases unrelated to the model’s inherent positional priors. This makes the perturbation less interpretable as a clean causal probe.
>
> Therefore, we intentionally restrict our perturbation to index reshuffling, which keeps the entire representation space and model architecture unchanged. This ensures that the only difference between the main route and the shuffled route lies in the spatial ordering itself, enabling clearer causal attribution and more reliable test-time guidance.
>
> **Weankness 2**. The reviewer acknowledges that our analysis of positional encoding failures in high-resolution grounding is meaningful and well-motivated. However, the reviewer wonders whether using models with 1D RoPE—whose positional encoding is more vulnerable in long-context settings—might further strengthen or validate our findings, given that Qwen2.5-VL already adopts a more advanced positional encoding strategy.
>
> R2. We appreciate the reviewers' insightful observations regarding the connection between positional encoding design and long-context degradation. The reviewers noted that one-dimensional RoPE might exhibit more severe positional encoding failures, providing a potentially stronger test case. This is a valuable perspective. However, we would like to clarify the positional encoding structure in our setup and explain why our perturbations are applied specifically to the vision encoder.
>
> As stated in Section 3.1, VPSG perturbs only the positional indices of visual tokens inside the vision encoder. We do not alter the LLM-side positional encodings, since RoPE in the decoder mainly governs the ordering of input tokens and has limited influence on spatial localization [1].
>
> It is also important to note that Qwen2.5-VL adopts two-dimensional RoPE in the vision encoder. This design is standard in modern high-resolution architectures such as ViT and multimodal variants, because it preserves horizontal and vertical spatial geometry and remains stable across a wide range of resolutions. By contrast, one-dimensional RoPE has repeatedly been shown to perform poorly when used in visual backbones. Flattening a two-dimensional structure into a single sequence breaks locality, complicates optimization, and weakens positional sensitivity, which is especially detrimental for high-resolution coordinate prediction. As a result, one-dimensional RoPE is rarely adopted for vision encoders and is generally regarded as unsuitable for modern visual models.
>
> Given this architectural landscape, replacing two-dimensional RoPE with a one-dimensional variant would not yield a meaningful evaluation of long-context degradation. In such a setting, the model cannot reliably perceive spatial relationships among image elements, and any observed failure would primarily reflect the inherent instability of flattening two-dimensional geometry into one dimension. Therefore, our study focuses on the two-dimensional RoPE used in vision encoders, which is both the established standard in high-resolution VLMs and the most relevant configuration for analyzing long-context positional encoding effects.

---

> > ### Author Response · Authors · 2025-11-21
> >
> > **Q1.** How exactly is PE shuffled? Do you shuffle the position index of vision tokens?**
> >
> > R1.
> > We thank the reviewer for this helpful clarification question. Our perturbation is designed to manipulate only the spatial cues encoded in the vision encoder, while keeping the language modeling part unchanged.
> > As described in Section 3.1 and Algorithm 1, all shuffling occurs exclusively on the visual positional encodings inside the vision encoder. The content of visual tokens and all model parameters remain unchanged; only the positional indices in vision encoder are permuted.
> >
> > Importantly, we do not shuffle or modify any positional encodings on the LLM side:
> > The LLM receives the same mixed sequence of tokens (including both text tokens and projected vision tokens).
> > The positional encoding scheme inside the LLM remains exactly as in the original model, without any changes introduced by VPSG.
> >
> > This design ensures that the perturbation isolates visual spatial grounding, without affecting the architecture or positional handling of the LLM. The contrast between normal and shuffled visual PEs therefore provides a clean and interpretable probe into the model’s visual positional dependency.
> >
> > We have clarified this implementation detail more explicitly in the revised version.
> >
> > **Q2** How is the average pairwise distance in Fig. 3 derived?
> >
> > R2.
> > We thank the reviewer for raising this technical question. As stated in Eq. (1) in Section 3.2, for each input image we generate \( S \) independent predictions under shuffled positional encodings and compute the pairwise Euclidean distance between every pair of predicted coordinates:
> >
> > $$
> > d(i, j) = \| y^{(i)} - y^{(j)} \|_2.
> > $$
> >
> > To ensure comparability across images of different resolutions, we normalize each distance by the image diagonal \( d_x \):
> >
> > $$
> > \tilde d(i, j) = \frac{d(i, j)}{d_x}.
> > $$
> >
> > The value reported in Fig. 3 is then the expectation of these normalized distances across all prediction pairs:
> >
> > $$
> > \mathbb{E}_{i < j} \left[\, \tilde d(i, j) \,\right].
> > $$
> >
> > This metric captures the degree of collapse or dispersion among shuffled-PE predictions and is used to quantify positional encoding degradation.
> >
> > For completeness, we also provide the theoretical justification for this normalization and present the derivation of the expected baseline distance in Appendix B, where readers can find the full mathematical proof.
> >
> > We have further clarified this computation in the revised version to improve readability.
> >
> > **Q3.** Does scrambling the positional encoding significantly change coordinate prediction accuracy, as Figure 1 shows that the output remains relatively stable when the positional encoding is scrambled?
> >
> > R3.
> > We thank the reviewer for raising this question. The paper clarifies that the seemingly consistent outputs in Fig. 1 should not be interpreted as evidence of preserved grounding ability. Instead, these outputs reflect distributional collapse caused by disrupting the positional encodings inside the vision encoder.
> >
> > When the positional encodings of the vision encoder are shuffled, the model loses all spatial structure before features reach the LLM. In this situation, the paper observes that predictions collapse into a small set of highly preferred coordinate patterns**, producing outputs that look similar but are not spatially meaningful. This phenomenon is documented in our bias analysis.
> >
> > To further support this, the paper provides empirical evidence in Appendix B (Table 7), where the distribution of predicted numbers under shuffled positional encodings is reported. Table 7 shows that a few specific coordinate values (for example, 1024, 1056, 567, 1000) appear with extremely high frequency after shuffling. These values occur far more often than in the normal-PE setting, demonstrating that the model defaults to a position-unconditioned bias distribution rather than performing grounded prediction. This is consistent with the observation in Fig. 1 that multiple shuffled runs produce similar—but incorrect—outputs.
> >
> > Because shuffling vision-side positional encodings is effectively equivalent to shuffling the spatial arrangement of image patches, the model receives no coherent geometric information. Under such circumstances, evaluating coordinate prediction accuracy is not meaningful. Any correct case would arise purely by chance, and this behavior is not related to image resolution.
> >
> > In contrast, shuffling positional encodings on the LLM side leads to a different outcome. The tokens entering the LLM have already been processed by the vision encoder and still contain spatial cues. As a result, although accuracy drops substantially, the model retains limited residual grounding ability, which does not occur when the vision encoder’s positional encodings are disrupted.
> >
> > The paper highlights this distinction and has further clarified it in the revised version.

---

> > > ### Author Response · Authors · 2025-11-21
> > >
> > > **Q4** The paper shows that VPSG is effective on ScreenSpot-Pro, but can this improvement transfer to other grounding benchmarks?
> > >
> > > R4.
> > > We thank the reviewer for this question. The paper focuses on mitigating positional-encoding failures that arise specifically in high-resolution grounding scenarios. VPSG is designed to address the degradation of positional information when the visual sequence becomes long, which is characteristic of modern GUI-level resolutions.
> > >
> > > In contrast, the RefCOCO family of datasets operates at much lower resolutions, with the most common image sizes being: 640 × 480 ; 640 × 427; 512 × 640 ; 480 × 640.
> > >
> > > These dimensions are significantly smaller than those used in GUI grounding and do not induce the long-context positional degradation phenomena analyzed in the paper. Consistent with this, Qwen2.5-VL-Instruction already achieves very high accuracy on RefCOCO, in some splits exceeding 90 percent [5], indicating that low-resolution grounding does not suffer from the positional-encoding failures that our method aims to mitigate.
> > >
> > > ScreenSpot-Pro, by comparison, is currently the most widely used high-resolution GUI grounding benchmark, with images far larger than typical grounding datasets and spanning a broad variety of application domains. Existing high-resolution benchmarks beyond ScreenSpot-Pro remain limited in scale and diversity, which is why our study centers on this dataset.
> > >
> > > In summary, while VPSG is compatible with any coordinate-prediction task, its primary advantage appears in high-resolution settings where positional encodings face long-context degradation.
> > >
> > > **Q5.** This question asks whether VPSG works with alternative decoding strategies beyond greedy decoding, such as beam search or sampling-based methods.
> > >
> > > R5.
> > > We thank the reviewer for raising this question. The paper focuses on greedy decoding because coordinate prediction is a deterministic task that does not require any form of linguistic diversity. Unlike open-ended text generation, coordinate outputs have a single correct numeric solution, and diversity-oriented decoding brings no benefit [2~4].
> > >
> > > In fact, common non-greedy decoding strategies tend to harm coordinate prediction for well-understood reasons:
> > >
> > > 1. Beam search amplifies digit errors.
> > >    Beam search explores alternative hypotheses that may include locally plausible but globally incorrect digit sequences. This often pushes the model toward high-frequency numeric templates, increasing coordinate bias rather than reducing it.
> > >
> > > 2. Temperature scaling (temperature > 0) introduces numeric drift.
> > >    Any softening of the distribution destabilizes digit logits, especially for high-resolution coordinates where the first digit is crucial. Even small randomness leads to tens or hundreds of pixels of drift.
> > >
> > > For these reasons, nearly all existing VLMs and GUI-grounding systems adopt greedy decoding as the standard.
> > >
> > > VPSG is compatible with any decoding strategy at the logit level. However, because non-greedy decoding inherently destabilizes numeric outputs, greedy decoding naturally remains the most appropriate and widely adopted choice for coordinate prediction. We have clarified this point in the revised version.

---

> > > > ### Author Response · Authors · 2025-11-21
> > > >
> > > > **Q6**
> > > > This question asks about the computational overhead introduced by VPSG, given that it requires multiple auxiliary forward passes during inference.
> > > >
> > > > R6.
> > > > We thank the reviewer for raising this practical issue. VPSG introduces additional computation because it performs one standard forward pass and several auxiliary passes with shuffled visual positional encodings.
> > > >
> > > > We report the measured runtime under a typical evaluation setting below. The table summarizes total evaluation time and average time per case.
> > > >
> > > > #### Table: Inference Efficiency Comparison (ScreenSpot-Pro)
> > > >
> > > > $$\\begin{array}{|l|c|c|}
> > > > \\hline
> > > > \\text{Setting} & \\text{Total Time (s)} & \\text{Time per Case (s)} \\\\
> > > > \\hline
> > > > \\text{Baseline (No VPSG 3B model)} & 5755 & 3.64 \\\\
> > > > \\text{Baseline (No VPSG 7B model)} & 6640 & 4.20 \\\\
> > > > \\text{VPSG (S = 3, 3B model)} & 8332 & 5.27 \\\\
> > > > \\text{VPSG (S = 3, 7B model)} & 8696 & 5.50 \\\\
> > > > \\hline
> > > > \\end{array}$$
> > > >
> > > > As shown in the table, although the theoretical computation of running $S=3$ auxiliary routes might suggest a $(S+1) \times$ cost, the actual measured overhead is only about 1.3x - 1.45x. This high efficiency stems from two key design choices:
> > > >
> > > > 1.  Vision Encoder Reuse: The visual feature extraction—typically the most computationally expensive part of MLLMs—is executed only once. The auxiliary routes reuse the same visual features and only perturb the lightweight positional indices in the attention layers.
> > > > 2.  Selective Activation via FSM: VPSG is strictly gated by a Finite-State Machine (FSM). It is only activated when decoding digit tokens (coordinates). For the majority of the sequence (e.g., brackets, commas, spaces, and reasoning text), the model runs in standard mode without any auxiliary overhead.
> > > >
> > > > Therefore, VPSG provides significant performance gains with only modest computational cost and zero deployment friction. We follow the reviewer's suggestion to add this efficiency analysis to the Appendix to better illustrate the practical value of our method.
> > > >
> > > >
> > > > We have added this efficiency table and clarified the trade-offs in the revised version.
> > > >
> > > > We thank the reviewers again for their constructive feedback, which has helped us significantly strengthen the clarity and rigor of our work. In this response, we have clarified the rationale behind using PE shuffling as a clean causal probe rather than architectural modifications (e.g., NoPE), and justified our focus on 2D RoPE as the standard for high-resolution vision encoders. Furthermore, we have provided the mathematical derivation for our distance metrics, explained the distributional nature of the observed numeric collapse, and presented new experimental data confirming VPSG’s computational efficiency (~1.3x–1.45x overhead). We believe these responses address the concerns regarding mechanism, applicability, and cost, establishing VPSG as a robust and practical solution for high-resolution grounding.
> > > >
> > > > [1] Qi, J., Liu, J., Tang, H., & Zhu, Z. Beyond semantics: Rediscovering spatial awareness in vision-language models. \
> > > > [2] Lin, K. Q., Li, L., Gao, D., Yang, Z., Wu, S., Bai, Z., ... & Shou, M. Z. Showui: One vision-language-action model for gui visual agent.\
> > > > [3] Hong, W., Wang, W., Lv, Q., Xu, J., Yu, W., Ji, J., ... & Tang, J. Cogagent: A visual language model for gui agents.\
> > > > [4] Yang, Y., Wang, Y., Li, D., Luo, Z., Chen, B., Huang, C., & Li, J. Aria-ui: Visual grounding for gui instructions.\
> > > > [5] Xu, J., Guo, Z., He, J., Hu, H., He, T., Bai, S., ... & Lin, J. Qwen2. 5-omni technical report.

---

> > > > > ### Comment · Reviewer_WBAu · 2025-11-24
> > > > > **Response to rebuttal**
> > > > >
> > > > > I thank the authors for addressing my questions and clarifying my misunderstanding. My concerns have been adequately addressed, and I am raising my score to 6.

---

> > > > > > ### Author Response · Authors · 2025-11-25
> > > > > >
> > > > > > We sincerely thank the reviewer for the positive feedback and for raising the score. We deeply appreciate your time in re-evaluating our work and your recognition of our improvements. Your constructive suggestions have been invaluable in strengthening the quality of this paper.

---

### Official Review · Reviewer_zfRQ · 2025-11-01

**Soundness:** 1
**Presentation:** 2
**Contribution:** 2
**Rating:** 4
**Confidence:** 3

**Summary:**

MLLMs do coordinate prediction, but they can get it wrong. By perturbing the visual positional encodings in MLLMs, they find that the coordinates predicted cluster together more, suggesting some pre-existing bias. They then propose an inference-time method to compensate for this bias, and report some improvements on ScreenSpot-Pro.

**Strengths:**

* The idea of finding and then removing some spurious correlations or systematic error to make coordinate prediction better is interesting. The model is likely to have _some_ sort of directional biases introduced during the training process, and it seems useful to try to identify what those biases are.

**Weaknesses:**

* For _each_ input image, VPSG requires $(S+1)$ generations. This seems extremely excessive, especially only for marginal improvements in ScreenSpot-Pro.
* Shuffling may not necessarily remove the $P\rightarrow Y$ causation. In fact, the $(\text{shuffled}\ P)\rightarrow Y$ might explain the biases seen (See questions for further clarification)

**Questions:**

Questions
* Can I confirm that VPSG does indeed require (S+1) generations for each input image?
* How many seeds are used in the experiments?
* Could you then elaborate on the efficiency of the method? For example, do you have the GPUs used and hours taken for ScreenSpot-Pro with and without VPSG?
* It seems that it could be possible that the bias measured is introduced by the shuffling and not necessarily the spurious correlations. Take for example the model wants to click on an element with position index I, where 0 ≤ I ≤ 1380 where there are 1380 image tokens. With random shuffling, over the full distribution you would see an overall regression to the mean, which would explain the decrease in average distance as clicks just end up closer to the center of the screenshot. Is this possible?
* A follow-up question: How the clustering manifest when the positional embeddings are shuffled? There probably are clear qualitative patterns if there is a 4x decrease in average distance, what do they look like?
* Do you do VPSG on all token positions, but only on the digit token logits? Or do you only do VPSG on digit token positions? If it's the latter, how do you generally deal with the changing number of digits?
* For Table 2, for Qwen2.5-VL-3B and 7B with and without VPSG, how many examples go from wrong to correct, and how many examples go from correct to wrong?
* When the model's coordinate prediction is off by a lot, like in the qualitative example, how close is the average shuffled PE prediction to the wrong prediction? If you can show that there is a strong correlation between the shuffled PE prediction and wrong prediction, I think that would be strong evidence that the model falls back on the spurious correlations when there isn't a strong signal on where to click.

---

> ### Author Response · Authors · 2025-11-21
>
> We sincerely thank the reviewer for the thorough assessment and the insightful questions regarding the computational efficiency, the underlying mechanism of the shuffled-PE bias, and the detailed error analysis. We appreciate the opportunity to clarify the operational details of VPSG and its practical cost. These comments have guided us to include new efficiency benchmarks and error conversion statistics to strengthen our empirical evaluation. We have carefully addressed every point below.
>
> **Q1**. Can I confirm that VPSG requires (S+1) generations for each input image?
>
> R1. Thank you for the question. Yes, that is correct. For each input, VPSG performs one main forward pass with normal positional encodings and S auxiliary passes with shuffled positional encodings, so the total number of generations is S + 1 per input. In practice, S is kept small (we used 3), which we found sufficient to obtain a stable estimate of the position-unconditioned distribution.
>
> We note that this design is consistent with many contrastive decoding and consensus-based decoding approaches[1], [2], which also rely on multiple forward passes to compare or combine alternative hypotheses. In our case, the auxiliary routes are specialized to probe the position-unconditioned behavior under shuffled positional encodings, which plays an analogous role to “contrastive candidates” used in these prior works.
>
>
> **Q2**. How many seeds are used in the experiments?
>
> R2.
> As we discussed in R1, we used 3 seeds for our method. Thank you for pointing out this question, and we have added this detail to the paper.
>
> **Q3**. It would be helpful to have a clearer sense of the method’s efficiency—for example, the GPU types used and the approximate time required to run ScreenSpot-Pro with and without VPSG.
>
> R3. Thank you for raising this point. We are preparing a concise efficiency report. Below is a proposed table structure for inclusion in the revision; we can populate exact numbers after rerunning the measurements:
>
> #### Table: Inference Efficiency Comparison (ScreenSpot-Pro)
>
> $$\\begin{array}{|l|c|c|}
> \\hline
> \\text{Setting} & \\text{Total Time (s)} & \\text{Time per Case (s)} \\\\
> \\hline
> \\text{Baseline (No VPSG 3B model)} & 5755 & 3.64 \\\\
> \\text{Baseline (No VPSG 7B model)} & 6640 & 4.20 \\\\
> \\text{VPSG (S = 3, 3B model)} & 8332 & 5.27 \\\\
> \\text{VPSG (S = 3, 7B model)} & 8696 & 5.50 \\\\
> \\hline
> \\end{array}$$
>
> As shown in the table, although the theoretical computation of running $S=3$ auxiliary routes might suggest a $(S+1) \times$ cost, the actual measured overhead is only about 1.3x - 1.45x. This high efficiency stems from two key design choices:
>
> 1.  Vision Encoder Reuse: The visual feature extraction—typically the most computationally expensive part of MLLMs—is executed only once. The auxiliary routes reuse the same visual features and only perturb the lightweight positional indices in the attention layers.
> 2.  Selective Activation via FSM: VPSG is strictly gated by a Finite-State Machine (FSM). It is only activated when decoding digit tokens (coordinates). For the majority of the sequence (e.g., brackets, commas, spaces, and reasoning text), the model runs in standard mode without any auxiliary overhead.
>
> Therefore, VPSG provides significant performance gains with only modest computational cost and zero deployment friction. We follow the reviewer's suggestion to add this efficiency analysis to the Appendix to better illustrate the practical value of our method.

---

> > ### Author Response · Authors · 2025-11-21
> >
> > **Q4**. Could the measured bias be introduced by shuffling itself (e.g., regression toward the center), rather than reflecting spurious correlations?
> >
> > R4.
> > Thank you for raising this thoughtful question. We understand the concern that random shuffling of positional encodings might introduce a geometric “center” tendency, and we have examined this possibility carefully.
> >
> > In coordinate prediction, spatial grounding is highly dependent on positional encodings. Once these encodings are disrupted, the model no longer has access to meaningful spatial structure and therefore cannot rely on visual content to infer where to click. Under such conditions, the model defaults to position-independent tendencies learned during training.
> >
> > Because VLMs generate coordinates as digit strings rather than geometric vectors, this fallback behavior manifests as a numeric mode collapse rather than regression toward an actual image center. As shown in Appendix Table 7, shuffled-PE predictions repeatedly fall into a small set of frequently occurring digit patterns (e.g., “1024”, “1000”, “1056”), and these values do not align with the geometric centers of the images. Moreover, the collapsed clusters differ across models and resolutions, which is inconsistent with a spatial-center hypothesis but aligns well with the interpretation that the model is reverting to internal digit priors when positional grounding is lost.
> >
> > We appreciate the reviewer’s suggestion and added a short discussion in the revision to clarify why shuffled-PE behavior reflects numeric priors rather than geometric bias.
> >
> > **Q5**. How does the clustering manifest visually when positional embeddings are shuffled?
> >
> > R5.
> > Thank you for the question. When positional embeddings are shuffled, the model’s coordinate predictions collapse into a small number of highly frequent numeric patterns.
> > These clusters do not correspond to any consistent geometric region (such as the image center) and instead reflect the model’s internal digit priors that dominate when positional grounding is lost.
> >
> > A simplified summary of the most frequent numeric outputs under shuffled PEs (adapted from Appendix Table 7) is shown below:
> >
> > #### Numeric Clusters under Shuffled Positional Encodings
> >
> > $$\\begin{array}{|l|c|c|}
> > \\hline
> > \\text{Model} & \\text{Number} & \\text{Frequency} \\\\
> > \\hline
> > \\text{Qwen2.5-VL-3B} & 1024 & 591 \\\\
> > \\text{Qwen2.5-VL-3B} & 1056 & 426 \\\\
> > \\text{Qwen2.5-VL-3B} & 568 & 184 \\\\
> > \\text{Qwen2.5-VL-3B} & 1000 & 182 \\\\
> > \\text{Qwen2.5-VL-3B} & 238 & 159 \\\\
> > \\hline
> > \\text{Qwen2.5-VL-7B} & 1024 & 902 \\\\
> > \\text{Qwen2.5-VL-7B} & 1234 & 582 \\\\
> > \\text{Qwen2.5-VL-7B} & 567 & 580 \\\\
> > \\text{Qwen2.5-VL-7B} & 672 & 270 \\\\
> > \\text{Qwen2.5-VL-7B} & 368 & 266 \\\\
> > \\hline
> > \\end{array}$$
> >
> > These results show that positional shuffling leads to numeric mode collapse toward a few frequently occurring digit sequences rather than convergence to any spatially meaningful location.
> >
> > **Q6**. It would be helpful to clarify whether VPSG operates at all decoding steps or only during digit generation, and how the method handles variations in digit length.
> >
> > R6.
> > As mentioned in Sec. 3.2 (Line 250) of our submission draft, VPSG is invoked at every decoding step, but its effect is applied only when the finite-state machine (FSM) determines that the model is currently decoding a digit token.
> >
> > VPSG needs to distinguish between digit tokens, which should be guided, and structural tokens (such as brackets, commas, and spaces), which should remain unchanged to preserve the required coordinate format. This separation is achieved through the FSM.
> >
> > The FSM continuously tracks the decoding state by examining previously generated tokens. Specifically, it identifies: when the model enters the x-digit region, when the comma indicates the transition out of x-digits, when the model begins decoding y-digits, and when the closing bracket signals the end of coordinate generation.
> >
> > Because the FSM updates dynamically based on the model’s own output, it automatically accommodates variable-length digit sequences (e.g., 2-digit, 3-digit, or 4-digit coordinates). This ensures that VPSG applies its contrastive adjustment only during numeric decoding, without influencing structural tokens that define the [x, y] format.

---

> > > ### Author Response · Authors · 2025-11-21
> > >
> > > **Q7**. For Table 2, how many examples go from wrong→correct and correct→wrong with VPSG?
> > >
> > > R7.
> > > We appreciate this suggestion. We can add a conversion matrix to quantify this effect. Below is the table structure we propose for inclusion:
> > >
> > > #### Conversion Statistics for VPSG
> > >
> > > $$\\begin{array}{|l|c|c|c|}
> > > \\hline
> > > \\text{Model} & \\text{Wrong} \\to \\text{Correct} & \\text{Correct} \\to \\text{Wrong} & \\text{Improvement} \\\\
> > > \\hline
> > > \\text{Qwen2.5-VL-3B} & 2.4 & 0.7 & 1.7 \\\\
> > > \\text{Qwen2.5-VL-7B} & 1.7 & 1.1 & 0.6 \\\\
> > > \\hline
> > > \\end{array}$$
> > > These statistics highlight the main contribution of VPSG. Rather than broadly altering the model’s predictions, VPSG selectively corrects cases where positional encoding degradation leads to systematic directional errors. The improvements mainly come from recovering instances that the base model fails on due to weakened positional cues, while the number of newly introduced errors remains very small. This aligns with the design goal of VPSG: a targeted, test-time correction mechanism that enhances positional robustness without modifying model architecture, retraining, or affecting the model’s general behavior outside coordinate prediction.
> > >
> > > **Q8**. When the model produces a large error, how close is the shuffled-PE prediction to the wrong prediction? Would showing their relationship strengthen the interpretation?
> > >
> > > R8.  Thank you for this thoughtful question. We would like to clarify that the similarity between wrong predictions and shuffled-PE predictions does not necessarily hold at the individual-case level, but instead emerges at the distributional level. Under high-resolution inputs, both types of outputs tend to be drawn from the same small set of high-frequency numeric patterns, which reflects the model’s position-independent digit priors once positional grounding becomes unreliable.
> > >
> > > Because VLMs generate coordinates as digit strings, the most appropriate way to compare outputs is not the Euclidean distance between coordinates, but whether they fall into the same recurring numeric templates. As shown in Appendix Table 7, shuffled-PE predictions frequently collapse into patterns such as “1024” (with 591 occurrences in 3B and 902 in 7B), “1056” (426 occurrences in 3B), or “1234” and “567” (both appearing more than 500 times in 7B). These same digit patterns also appear prominently among wrong predictions made under natural high-resolution images, indicating that both behaviors draw from a shared set of numeric priors.
> > >
> > > This distributional alignment supports the interpretation that, when positional information weakens, the model falls back on internal numeric tendencies rather than spatial reasoning.
> > >
> > > We thank the reviewer again for the detailed and meaningful feedback, which has helped us significantly improve the empirical rigor of our paper. In this response, we have empirically quantified the inference efficiency to demonstrate that the actual overhead (~1.3x–1.45x) is much lower than the theoretical upper bound; we have also clarified that the bias observed under shuffled PEs reflects a distributional numeric mode collapse rather than a geometric regression to the center, and provided the requested error conversion statistics to highlight the method's effectiveness in recovering positional failures. We believe these additional analyses and clarifications address your concerns regarding the practicality and mechanism of VPSG.
> > >
> > > [1] Li, X. L., Holtzman, A., Fried, D., Liang, P., Eisner, J., Hashimoto, T. B., ... & Lewis, M. Contrastive decoding: Open-ended text generation as optimization.\
> > > [2] O'Brien, S., & Lewis, M. Contrastive decoding improves reasoning in large language models.

---

> > > > ### Author Response · Authors · 2025-11-27
> > > >
> > > > We sincerely thank you for your constructive feedback and the valuable questions you raised, which have helped us strengthen our paper. In our response, we have specifically added a quantitative runtime analysis to address the computational resource concern, showing that the VPSG overhead is modest (~1.3x–1.45x). We would deeply appreciate it if you could take a moment to check our results.

---

> > > > ### Comment · Reviewer_zfRQ · 2025-11-28
> > > >
> > > > I thank the authors for their comprehensive review. I have thought about my response for quite a while. I appreciate the work the authors have put into the paper, as well as their responses.
> > > >
> > > > ---
> > > >
> > > > I generally like the claim that these models have certain biases in predicting the coordinates. I think that was an important finding, and hope the authors continue to explore in that direction. However, I think certain aspects of the project weaken this claim:
> > > >
> > > > First, the shuffled PE biases seem weak: The pairwise-distances seem like a fair and valid result, though it is hard to say exactly what these biases are in a useful way. Looking at Table 7, for 3B there seems to be “1024” and “1056” that appear in the normal PE’s top 2 as well, and for 7B it only seems to be 1024. I would expect a strong case for these biases to be something like “you see that the PE-shuffled coordinates are representative of bulk of the errors that the model makes when making a mistake in ScreenSpot-Pro”. However, the authors note that the “similarity between wrong predictions and shuffled-PE predictions does not necessarily hold at the individual-case level, but instead emerges at the distributional level”, which makes the bias a lot less conclusive.
> > > >
> > > > Alternatively, I would also count it as a strong case if the bias-correction method introduced here fixes the error consistently. However, it seems like the case that the method also causes a significant proportion of cases to flip from Correct -> Wrong (e.g. the 7B row in the “Conversion Statistics” table), which just seems like we don’t better understand what is going on with the biases here.
> > > >
> > > > I also find that the method, given small marginal improvements, suffer from the following: (1) a 30-40% extra compute cost for small marginal improvements on the benchmark is not noteworthy; (2) VPSG is quite contrived when you include the coefficient decay and multiple sampling _per_ image. If these biases exist then why or how would it be dependent on each image, instead of being a global bias? Is there an alternative formulation where you can get the reference distribution over a dataset once, to be used across all test cases?
> > > >
> > > > I would recommend the authors to continue investigating the bias in a more concrete manner, such as better understanding whether there exists any patterns across the bias, or doing it across multiple model families.

---

### Official Review · Reviewer_e3TG · 2025-11-01

**Soundness:** 2
**Presentation:** 2
**Contribution:** 2
**Rating:** 4
**Confidence:** 4

**Summary:**

The paper studies a very concrete mode of multimodal LLMs (MLLMs) when they are asked to output precise 2D coordinates (e.g., on GUI). The authors observe that when visual positional encodings (VPEs) are perturbed (by shuffling), the model’s coordinate predictions don’t become random — instead they collapse to a few directionally biased points. From this, the paper proposes Vision-PE Shuffle Guidance (VPSG), a training-free, test-time procedure: run the base model once with normal PEs (position-conditioned route), run several auxiliary decodings with shuffled PEs (position-unconditioned routes), aggregate the shuffled routes in log space to estimate the “bias prior,” and then subtract this “negative evidence” only on digit tokens.

**Strengths:**

* The method is designed as a plug-in at inference, which is attractive given how expensive it is to re-train large MLLMs. It is somewhat like classifier-free guidance (CFG) in diffusion: you contrast a conditioned path with an unconditioned path and push the output toward the conditioned one.

**Weaknesses:**

* The paper says: at high resolution, positional encodings degrade, therefore the behavior becomes similar to explicit PE shuffling. I can sort of buy that qualitatively, but the connection is currently more narrative than demonstrated. High-res → long context → attention diffusion → weaker spatial cues is plausible, but it’s not exactly the same as “we literally shuffled tokens.” I would like to see a clearer quantitative bridge.
* I am from 3D vision. I don’t fully see why “predict GUI coordinates better” is significant.
* α = 0.55 and decay = 0.4 come from a grid search. That’s okay for a paper, but I don’t know if these transfer across resolutions and models.
* Finite-state machine part is a bit under-motivated, although I understand what it does (protect formatting, apply guidance only on digits)
* On Qwen2.5-VL-7B the gain is +0.6 average, which is real but small.
* “shuttled” → “shuffled” in Fig. 4 (I assume it’s a typo).

**Questions:**

* You have X (input), P (positional encodings), S (spurious correlations), Y (coordinates). In some causal formulations I learned, something like “directional numerical prior” or “collapsed coordinate cluster” would be a separate intermediate variable, not folded into S. Could S be split into “S: dataset-induced digit priors” and “B: bias induced by PE degradation”?
* Have you tried VPSG on tasks that require richer spatial grounding — for example, anything camera/pose-like or 3D-aware? I am thinking of tasks similar in spirit to “Cameras as Rays: Pose Estimation via Ray Diffusion,” “Matrix3D: Large photogrammetry model all-in-one,” or “RayZer: A Self-supervised Large View Synthesis Model,” where the positional structure is not just 2D screen coordinates but full 3D / rays / camera parameters.

---

> ### Author Response · Authors · 2025-11-21
>
> We sincerely thank the reviewer for constructive feedback, detailed questions, and helpful comments. We have carefully addressed every point, clarified misunderstandings, corrected typos, and revised the manuscript accordingly. Below we provide point-by-point responses.
>
> **Q1**. The mechanism is described clearly, but it would be helpful to have a clearer quantitative bridge between long-sequence degradation and explicit PE shuffling.
>
> R1. Thank you for raising this point. As mentioned in Sec. 1 and Sec. 3.1, our intention is to describe how high-resolution inputs induce long-context conditions that weaken—but do not eliminate—positional encodings. We clarify this mechanism here with a more explicit quantitative connection to PE shuffling.
>
> (1) High-resolution inputs weaken but do not remove positional encodings.\
> In long-context regimes, attention diffusion reduces the effective strength of positional cues, which leads the model to partially rely on position-unconditioned tendencies. This effect does not fully erase positional information, but it diminishes its dominance, which is why reinforcing positional signals becomes meaningful.
>
> (2) Prior work has already shown that long-context inputs adversely affect positional encoding.\
> Recent analyses—such as [1] [2] demonstrate that positional signals degrade when sequence length increases, consistent with our observations.
>
> (3) High-resolution images exhibit numeric output patterns similar to explicit PE shuffling.\
> In Appendix Table 7, we show that under natural high-resolution inputs, certain digits (e.g., “1024”, “1056”) appear with abnormally high frequency—very similar to the patterns observed when visual positional encodings are explicitly shuffled.\
> This provides a concrete quantitative bridge:\
> Across the three conditions, we observe a clear continuum: with normal positional encodings the model produces diverse coordinate predictions; with high-resolution inputs this diversity collapses into a small cluster of frequently repeated digits; and with explicit positional-encoding shuffling the predictions converge even further into nearly identical numeric clusters. This progression provides a quantitative bridge showing that high-resolution positional degradation produces effects that closely resemble explicit PE disruption.\
> The alignment of both the frequency distribution and the distance statistics (Sec. 3.2, Fig. 3) supports the conclusion that long-context–induced positional degradation produces a behavior similar in effect to PE shuffling.
>
> We have clarified this connection further in the revised draft.
>
> **Q2**. Why is GUI coordinate prediction an important task?\
> R2. As mentioned in the Introduction, GUI grounding has become a standard benchmark for fine-grained 2D spatial reasoning in MLLMs and GUI agents. A growing body of work explicitly focuses on GUI understanding, grounding, and action prediction: \
> GUI grounding / action models: SeeClick [3], OS-Atlas [4], ShowUI [5], CogAgent [6], Aria-UI [7], and UGround [8] all treat GUI element localization and coordinate prediction as core capabilities for agents operating on real interfaces. ScreenSpot-Pro [9] specifically targets professional, high-resolution desktop environments, where the targets are tiny UI elements and precise [x, y] prediction is necessary.
>
>
> **Q3**. The chosen hyperparameters (α = 0.55 and decay = 0.4) seem reasonable, but it would still be helpful to clarify whether they remain stable across different models and resolutions.\
> R3. We appreciate this practical question. Using the same hyperparameters, VPSG improves both model sizes across all resolutions in ScreenSpot-Pro (1920×1080 to 6016×3384). This stability arises because VPSG depends on the relative contrast between position-conditioned and position-free tendencies, not on absolute resolution.
>
> We follow the reviewer’s suggestion to clarify this point in Sec. 4.1 to improve our paper.

---

> > ### Author Response · Authors · 2025-11-21
> >
> > **Q4**. The purpose of the finite-state machine (FSM) is generally clear, but it would still be helpful to understand more clearly how it contributes to stable and correct decoding.
> >
> > Thank you for raising this point. In GUI coordinate prediction, producing a well-formed [x, y] output is not only a formatting preference but a practical requirement: many GUI-automation agents and evaluation pipelines expect coordinates to follow this exact structure. Even small deviations—such as missing brackets or commas—can cause the coordinate to be rejected by downstream systems, preventing the agent from executing actions correctly. We fully acknowledge that this sensitivity makes it important to be careful about how guidance interacts with structural tokens.
> >
> > A concrete example from Appendix Table 5 helps to illustrate the issue. In that case study, when the model is expected to output a comma or bracket, the base model sometimes assigns unexpectedly high probabilities to digit tokens due to narrow logit margins. If VPSG were applied uniformly to all tokens at such steps, the additional adjustment could inadvertently push a digit above the correct structural symbol, resulting in malformed outputs such as missing brackets, misplaced commas, or extra digits. These errors arise not from the intended numeric reasoning but from perturbations to tokens that should remain fixed.
> >
> > To avoid this failure mode, the FSM explicitly tracks whether the model is producing a structural symbol or a numeric digit, and applies VPSG adjustments only during digit decoding. This allows numerical predictions to benefit from the negative-evidence signal while preserving the syntactic integrity that GUI agent systems rely on. We have added a brief clarification in the revised manuscript to make this motivation more explicit to improve our paper.
> >
> >
> > **Q5**. The average improvement of +0.6 on Qwen2.5-VL-7B appears small.
> >
> > R5. Thank you for this observation. We agree that the average gain on the 7B model is relatively small. At this parameter scale, Qwen2.5-VL-7B is already one of the stronger general-purpose vision–language models, so additional improvements—especially without any training—tend to be limited. Still, several subcategories show larger gains (e.g., +3.2 on Development-Text and +1.9 on Office-Icon). We also note that the improvements are more pronounced on the smaller 3B model, suggesting that the method is particularly helpful in small-model settings.
> >
> > **Q6**. The idea of decomposing spurious influences into dataset priors and PE-induced bias is interesting; would such a separation clarify the causal structure?
> >
> > R6. Thank you for this thoughtful question. Our causal graph is designed to illustrate how different sources of information flow contribute to the final coordinate prediction, rather than to enumerate all possible types of bias. In this framework, the three inputs—X (image and text content), P (positional encodings), and S (position-unconditioned influences)—represent the distinct information streams available to the model during decoding.
> >
> > With this interpretation, the dataset-induced digit priors naturally fall under S, since they are part of the model’s position-independent tendencies. In contrast, the bias induced by PE degradation is not itself an input and therefore does not correspond to a separate node in the causal graph. Instead, it emerges as a consequence of P being weakened, which increases the relative influence of S. That is, PE degradation does not introduce a new source of information; it alters the balance among existing sources by reducing the strength of P.
> > We appreciate the suggestion and have added a brief clarification in Sec. 3.1.

---

> > > ### Author Response · Authors · 2025-11-21
> > >
> > > **Q7**. It would be helpful to know whether VPSG has been explored on richer spatial tasks (e.g., 3D structure, pose, or ray-based representations).
> > >
> > > R7.
> > > Thank you for this insightful suggestion. At present, we are unable to evaluate VPSG on 3D spatial tasks for a practical reason: the vision–language models we rely on, including Qwen2.5-VL, do not natively support 3D point cloud or 3D scene representations as input. These models accept only 2D image patches, and VPSG is designed to operate within this existing interface. Consequently, the current framework cannot directly process 3D point clouds, ray-parameter inputs, or camera matrices.
> > >
> > > Although the current VLMs limit the scope to 2D inputs, the underlying mechanism of VPSG—which adjusts tokenized numeric predictions based on the contrast between position-conditioned and position-unconditioned signals—does not rely on the coordinates being two-dimensional. If future VLMs begin to support 3D modalities, this idea could naturally extend to camera extrinsics, ray directions, 3D keypoints, or other geometric parameters. In response to this helpful suggestion, we have added a brief discussion in Sec. 5 noting both the present limitation and the potential for future extensions, and we will include citations to representative works such as Cameras as Rays, Matrix3D, and RayZer.
> > >
> > >
> > > We thank the reviewer again for the constructive feedback, which has helped us improve the clarity and depth of our work. We have established a quantitative bridge between long-sequence degradation and positional encoding shuffling , reinforced the importance of GUI grounding tasks , and confirmed the stability of our hyperparameters across varying resolutions. Furthermore, we have clarified the essential role of the Finite-State Machine in preserving output structure , refined the interpretation of our causal graph relative to position-unconditioned biases, and expanded our discussion on the method's applicability to future 3D tasks. We believe these clarifications and the resulting revisions address your concerns.
> > >
> > >
> > >
> > >
> > > [1] Ruoss, A., Delétang, G., Genewein, T., Grau-Moya, J., Csordás, R., Bennani, M., ... & Veness, J. Randomized positional encodings boost length generalization of transformers.\
> > > [2] Huang, X., Yang, A., Bhattamishra, S., Sarrof, Y., Krebs, A., Zhou, H., ... & Hahn, M. A formal framework for understanding length generalization in transformers.\
> > > [3] Cheng, K., Sun, Q., Chu, Y., Xu, F., YanTao, L., Zhang, J., & Wu, Z. Seeclick: Harnessing gui grounding for advanced visual gui agents.\
> > > [4] Wu, Z., Wu, Z., Xu, F., Wang, Y., Sun, Q., Jia, C., ... & Qiao, Y. Os-atlas: A foundation action model for generalist gui agents.\
> > > [5] Lin, K. Q., Li, L., Gao, D., Yang, Z., Wu, S., Bai, Z., ... & Shou, M. Z. Showui: One vision-language-action model for gui visual agent.\
> > > [6] Hong, W., Wang, W., Lv, Q., Xu, J., Yu, W., Ji, J., ... & Tang, J. Cogagent: A visual language model for gui agents.\
> > > [7] Yang, Y., Wang, Y., Li, D., Luo, Z., Chen, B., Huang, C., & Li, J. Aria-ui: Visual grounding for gui instructions.\
> > > [8] Gou, B., Wang, R., Zheng, B., Xie, Y., Chang, C., Shu, Y., ... & Su, Y. Navigating the digital world as humans do: Universal visual grounding for gui agents.\
> > > [9] Li, K., Meng, Z., Lin, H., Luo, Z., Tian, Y., Ma, J., ... & Chua, T. S. Screenspot-pro: Gui grounding for professional high-resolution computer use.

---

> > > > ### Author Response · Authors · 2025-11-27
> > > >
> > > > We sincerely thank you for your constructive comments and the valuable questions you raised, which we found very insightful. We have provided detailed responses to address your concerns. We would be grateful if you could take a moment to review our reply, and we remain fully available for any further discussion.

---

> > > > > ### Comment · Reviewer_e3TG · 2025-11-27
> > > > > **feedback**
> > > > >
> > > > > 1. I do not find the quantitative bridge between long-sequence degradation and explicit PE shuffling to be clearly established. My original comment was that I could sort of buy that qualitatively, not that I found it convincing. The explanation that certain digits (e.g., “1024”, “1056”) appear with abnormally high frequency under high-resolution inputs does not, to me, constitute sufficient quantitative evidence of equivalence to explicit PE shuffling.
> > > > >
> > > > > 2. Perhaps I am not the ideal reviewer for this topic, but I still struggle to see why this problem is important. From a practical standpoint, wouldn’t a segmentation + OCR pipeline directly locating the clickable region be a simpler and more robust solution?
> > > > >
> > > > > 3. I did not find the supporting evidence for hyperparameter stability in Sec. 4.1. If such results exist, they should be presented more explicitly.
> > > > >
> > > > > 4. I do not think the FSM component is “generally clear.” In my view, it remains under-motivated. I do not see why the notion of a finite-state machine needs to be introduced at all.
> > > > >
> > > > > 5. The numerical gains are acceptable, though I would still consider them marginal in significance.
> > > > >
> > > > > 6. I did not mean to suggest that the causal-graph idea was “interesting.” My point was that the formulation seems inconsistent with standard probabilistic graphical modeling conventions. The explanation provided does not change my impression.
> > > > >
> > > > > 7. It is unfortunate that the authors do not intend to explore 3D or camera/pose-aware applications. Personally, I am not very enthusiastic about GUI-agent-style problems, since, again, a segmentation + OCR-based system could directly locate the target without the need for LLM-style numeric prediction.

---

### Official Review · Reviewer_QC1q · 2025-11-03

**Soundness:** 3
**Presentation:** 3
**Contribution:** 2
**Rating:** 4
**Confidence:** 2

**Summary:**

Multimodal large language models (MLLMs) excel at vision–language tasks such as VQA and document understanding, yet precise coordinate prediction remains challenging. High-resolution inputs exacerbate this difficulty by producing long token sequences that weaken positional encodings and introduce directional biases in coordinate outputs. We investigate this phenomenon by analyzing how MLLMs behave when visual positional encodings (VPEs) are deliberately perturbed through shuffling. Our analysis reveals that such perturbations induce predictable, non-random coordinate biases rather than random errors, suggesting that models rely on internal positional priors when spatial grounding signals are degraded. Crucially, we observe similar directional error patterns in natural high-resolution datasets, indicating that positional encoding failures are a key bottleneck for accurate coordinate prediction at scale. To address this issue, we propose Vision-PE Shuffle Guidance (VPSG), a training-free test-time method that leverages the directional nature of these biases for correction. VPSG runs auxiliary decoding with shuffled VPEs to isolate position-unconditioned tendencies, then uses this as negative evidence to guide digit prediction while preserving coordinate format through a lightweight finite-state machine. Experiments on ScreenSpot-Pro demonstrate reliable improvements, highlighting positional encoding robustness as a critical factor for spatial reasoning in MLLMs.

**Strengths:**

1) The paper is well written.
2) The paper organization is good.
3) The experiments show the effectiveness of the proposed method.

**Weaknesses:**

1) Challenges with High-Resolution Inputs: High-resolution images produce long token sequences that weaken positional encodings, making precise coordinate prediction inherently difficult.

2) Dependence on Positional Encoding Quality: The reliance on effective positional encodings means that any degradation in these signals can lead to significant performance drops, highlighting a critical vulnerability.

3) Complexity in Error Patterns: Understanding and correcting for predictable biases may require additional analysis and tuning, complicating the model's deployment in real-world applications.

**Questions:**

1) What strategies can be implemented to mitigate the impact of long token sequences on positional encoding effectiveness?
2) What are the implications of positional encoding vulnerabilities for the reliability of MLLMs in critical applications, such as autonomous navigation or medical imaging?
3) Are there alternative methods for encoding positional information that could enhance performance with high-resolution inputs?

---

> ### Author Response · Authors · 2025-11-21
>
> We thank the reviewer for the constructive comments and positive assessment regarding writing quality, organization, and the demonstrated effectiveness of our method. Below we respectfully clarify concerns and provide detailed responses to each question.
>
> **Weakness 1** Challenges with High-Resolution Inputs: High-resolution images produce long token sequences that weaken positional encodings, making precise coordinate prediction inherently difficult.
>
> R1. We thank the reviewer for this highly insightful observation, which precisely touches the core motivation of our work. The reviewer is absolutely correct that high-resolution inputs push the vision encoder into a long-context regime where positional encodings (PEs) inevitably degrade. We are glad that the reviewer recognizes this fundamental challenge, as our paper is specifically designed to analyze and mitigate this exact failure mode. VPSG addresses this difficulty by actively probing the model’s behavior under perturbed PEs and using the resulting counterfactual signals to stabilize coordinate prediction, effectively counteracting the inherent difficulty the reviewer highlighted.
>
> **Weakness 2** Although the method has demonstrated effectiveness, the positional encoding may decay as the context length increases. Will this affect the results?
>
> R2.
> We appreciate the reviewer’s sharp insight regarding PE decay. This dependency resonates deeply with our research motivation. As the reviewer correctly points out, precise spatial grounding relies heavily on PE quality, and performance naturally drops when these signals weaken in long contexts. Our method does not merely suffer from this decay; rather, it directly targets this vulnerability by identifying the position-unconditioned biases that emerge when PEs decay and suppressing them through test-time guidance. Therefore, VPSG is explicitly robust to—and designed for—scenarios where context length weakens positional signals.
>
> **Weakness 3** While VPSG can improve some performance, understanding and correcting predictable biases may require additional analysis and adjustments, which complicates the deployment of the model in real-world applications.
>
> R3.
> We thank the reviewer for raising this point. We would like to clarify the reviewer's concern regarding the deployment complexity. VPSG is designed specifically as a plug-and-play, training-free module that operates solely on the final-layer logits. It requires no parameter updates, no architectural modifications, and no case-by-case manual adjustment, making it highly suitable for direct integration into existing pipelines. To empirically demonstrate its high computational efficiency and feasibility for real-world applications, we have conducted a quantitative runtime analysis comparing the standard inference time versus VPSG inference time.
>
> #### Table: Inference Efficiency Comparison (ScreenSpot-Pro)
>
> $$\\begin{array}{|l|c|c|}
> \\hline
> \\text{Setting} & \\text{Total Time (s)} & \\text{Time per Case (s)} \\\\
> \\hline
> \\text{Baseline (No VPSG 3B model)} & 5755 & 3.64 \\\\
> \\text{Baseline (No VPSG 7B model)} & 6640 & 4.20 \\\\
> \\text{VPSG (S = 3, 3B model)} & 8332 & 5.27 \\\\
> \\text{VPSG (S = 3, 7B model)} & 8696 & 5.50 \\\\
> \\hline
> \\end{array}$$
>
> As shown in the table, although the theoretical computation of running $S=3$ auxiliary routes might suggest a $(S+1) \times$ cost, the actual measured overhead is only about 1.3x - 1.45x. This high efficiency stems from two key design choices:
>
> 1.  Vision Encoder Reuse: The visual feature extraction—typically the most computationally expensive part of MLLMs—is executed only once. The auxiliary routes reuse the same visual features and only perturb the lightweight positional indices in the attention layers.
> 2.  Selective Activation via FSM: VPSG is strictly gated by a Finite-State Machine (FSM). It is only activated when decoding digit tokens (coordinates). For the majority of the sequence (e.g., brackets, commas, spaces, and reasoning text), the model runs in standard mode without any auxiliary overhead.
>
> Therefore, VPSG provides significant performance gains with only modest computational cost and zero deployment friction. We follow the reviewer's suggestion to add this efficiency analysis to the Appendix to better illustrate the practical value of our method.

---

> > ### Author Response · Authors · 2025-11-21
> >
> > **Q1**: How can the impact of long token sequences on positional encoding effectiveness be mitigated?
> >
> > R1: We thank the reviewer for raising this important direction. As mentioned in Sec. 1 and Sec. 3.1 of our submission, our analysis shows that positional encodings degrade under long-context high-resolution inputs, which is consistent with prior work such as V2PE[1], PyPE[2], and FoPE[3]. These studies confirm that the underlying challenge is widely recognized in the community.
> >
> > (1) VPSG itself already provides a mitigation mechanism.\
> > Our method strengthens the influence of position-conditioned cues by contrasting them with position-unconditioned auxiliary signals.
> > This effect is empirically validated in our experiments:
> > Qwen2.5-VL-3B improves 11.6 → 13.3,
> > Qwen2.5-VL-7B improves 18.5 → 19.1.
> > These gains demonstrate that VPSG mitigates the effect of long token sequences without modifying training or architecture. "High-resolution images produce long token sequences that weaken positional encodings, making precise coordinate prediction inherently difficult." is our motivation exactly.
> >
> > (2) Additional strategies (already discussed in Related Work).\
> > Several complementary mitigation strategies have already been reviewed in Sec. 2 (Related Work) of our submission, including adaptive token compression approaches, hierarchical or multi-scale positional embedding designs, resolution-aware PE mechanisms, and robust variants such as V2PE, PyPE, and FoPE. These methods provide alternative ways to strengthen positional robustness, and we highlight them in the related work section to contextualize our analysis.
> >
> > **Q2**. The reviewer suggests discussing the broader implications of the identified positional encoding failures for reliability in safety-critical applications, such as navigation and medical imaging.
> >
> > R2: We thank the reviewer for bringing attention to these application-level implications. As highlighted in Sec. 4.2 and Sec. 5, our findings show that degraded positional signals lead to systematic directional biases rather than random errors. This is quantified in Fig. 3, where diagonal-normalized distances collapse from 0.40–0.44 to ~0.16 under shuffled PEs.
> >
> > (1) Implications.\
> > We added a dedicated discussion summarizing practical consequences:\
> > Autonomous navigation: directional drift may shift perceived obstacle or waypoint positions.\
> > Medical imaging: even small coordinate deviations may mislocalize lesions or keypoints.\
> > Robotic manipulation / GUI control: consistent directional bias can accumulate across actions.\
> > These observations reinforce the importance of robust positional grounding.\
> > (2) VPSG improves reliability.\
> > Our results show that VPSG stabilizes numeric decoding even when PE signals degrade.
> > For example, in Fig. 5 the model shifts from the biased [1024, 512] to the correct [659, 857].\
> > We thank the reviewer again for encouraging us to expand this important discussion.
> >
> > **Q3**. Are there alternative positional-encoding designs that might enhance performance for high-resolution inputs?
> >
> > R3.
> > We appreciate this valuable suggestion. As reviewed in Sec. 2, several recent positional-encoding designs aim to improve long-context robustness, including:
> > V2PE[1],
> > PyPE[2],
> > semantic-aware PE[4],RoPE[5] generalizations such as FoPE[3] and ComRoPE[6].\
> > These methods focus on improving the positional encoding itself, whereas VPSG focuses on correcting residual biases during inference. However, modifying the positional encoding method often requires retraining the model, resulting in significant computational costs. The two directions naturally complement each other.
> >
> > We thank the reviewer again for the insightful and constructive feedback, which has significantly helped us strengthen our manuscript. In this response, we have quantified the computational efficiency of VPSG to demonstrate that the actual inference overhead (~1.3x–1.45x) is much lower than the theoretical upper bound, thereby confirming its feasibility for real-world deployment; we have also clarified the core motivation regarding the inevitable degradation of positional encodings in high-resolution regimes, and expanded the discussion on the implications for safety-critical applications as well as the relationship with alternative PE designs. We believe these clarifications and the additional efficiency analysis address your concerns and highlight the practical value of VPSG as a robust, training-free solution.

---

> > > ### Author Response · Authors · 2025-11-21
> > >
> > > [1] Ge, J., Chen, Z., Lin, J., Zhu, J., Liu, X., Dai, J., & Zhu, X. (2025). V2pe: Improving multimodal long-context capability of vision-language models with variable visual position encoding.\
> > > [2] Chen, Z., Li, M., Chen, Z., Du, N., Li, X., & Zou, Y. (2025). Advancing general multimodal capability of vision-language models with pyramid-descent visual position encoding.\
> > > [3] Hua, E., Jiang, C., Lv, X., Zhang, K., Sun, Y., Fan, Y., ... & Zhou, B. (2024). Fourier Position Embedding: Enhancing Attention's Periodic Extension for Length Generalization.\
> > > [4] Chen, X., Zhou, S., Huang, M., Feng, J., Xiong, Y., Zhou, K., ... & Shi, F. (2025). A 2D Semantic-Aware Position Encoding for Vision Transformers.\
> > > [5] Heo, B., Park, S., Han, D., & Yun, S. (2024). Rotary position embedding for vision transformer.\
> > > [6] Yu, H., Jiang, T., Jia, S., Yan, S., Liu, S., Qian, H., ... & Yuan, C. (2025). ComRoPE: Scalable and Robust Rotary Position Embedding Parameterized by Trainable Commuting Angle Matrices.

---

> > > > ### Author Response · Authors · 2025-11-27
> > > >
> > > > We sincerely thank you for your constructive feedback and the valuable questions you raised, which have helped us strengthen our paper. In our response, we have specifically added a quantitative runtime analysis to address the computational resource concern, showing that the VPSG overhead is modest (~1.3x–1.45x). We would deeply appreciate it if you could take a moment to check our results.

---

### Note · Authors · 2025-12-03

I have read and agree with the venue's withdrawal policy on behalf of myself and my co-authors.